# Can Nutrients and Dietary Supplements Potentially Improve Cognitive Performance Also in Esports?

**DOI:** 10.3390/healthcare10020186

**Published:** 2022-01-18

**Authors:** Monika Szot, Ewa Karpęcka-Gałka, Remigiusz Dróżdż, Barbara Frączek

**Affiliations:** 1Department of Sports Dietetics, Gdansk University of Physical Education and Sport, 80-336 Gdansk, Poland; szotmonika@hotmail.com; 2Doctoral School of Physical Culture Sciences, University School of Physical Education in Krakow, Jana Pawla II 78, 31-571 Cracow, Poland; karpeckaewa@gmail.com; 3Department of Economics and Business, Gdansk University of Physical Education and Sport, 80-336 Gdansk, Poland; remigiusz.drozdz@awf.gda.pl; 4Department of Sports Medicine and Human Nutrition, Institute of Biomedical Sciences, University School of Physical Education in Krakow, Jana Pawla II 78, 31-571 Cracow, Poland

**Keywords:** esports, cognitive performance, cognition, video game, esports athletes, nutrition habits

## Abstract

Factors influencing brain function and cognitive performance can be critical to athletic performance of esports athletes. This review aims to discuss the potential beneficial effects of micronutrients, i.e., vitamins, minerals and biologically active substances on cognitive functions of e-athletes. Minerals (iodine, zinc, iron, magnesium) and vitamins (B vitamins, vitamins E, D, and C) are significant factors that positively influence cognitive functions. Prevention of deficiencies of the listed ingredients and regular examinations can support cognitive processes. The beneficial effects of caffeine, creatine, and probiotics have been documented so far. There are many plant products, herbal extracts, or phytonutrients that have been shown to affect precognitive activity, but more research is needed. Beetroot juice and nootropics can also be essential nutrients for cognitive performance. For the sake of players’ eyesight, it would be useful to use lutein, which, in addition to improving vision and protecting against eye diseases, can also affect cognitive functions. In supporting the physical and mental abilities of e-athletes the base is a well-balanced diet with adequate hydration. There is a lack of sufficient evidence that has investigated the relationship between dietary effects and improved performance in esports. Therefore, there is a need for randomized controlled trials involving esports players.

## 1. Introduction

The increasingly frequent incidence of neurodegenerative diseases and the lack of an effective method for their treatment have resulted in an increased interest in dietary supplements to improve cognitive functions and prevent their decline [1]. A micronutrient deficit can lead to a decrease in cognitive abilities, which also deteriorate with age. Not only the elderly but also young and middle-aged people are exposed to micronutrient deficiencies due to their demanding lifestyles and insufficient diets [2]. Therefore, it is suggested that e-athletes may also be at risk.

In general opinion, esports (electronic sport) is a relatively new and rapidly growing sport involving players competing in video game tournaments. Although the genesis of e-tournaments was in the 1970s, the first professional competition started in 1997 [3]. Currently, this discipline is becoming more and more popular among players and viewers [4]. The global audience for esports competitions in 2019 had 443 million viewers, which translated into a market value of USD 68.2 billion [5]. It is estimated that the popularity of esports will grow in subsequent years [6]. Although women participate in esports, men continue to dominate the field [7]. Mainly due to the competitive nature of esports, but also due to the “physicality” and “skill” necessary to play at a high level, it was found that esports have characteristics similar to traditional sports. Additionally, the current organizational structure of esports and social interest in this discipline contributed to its acceptance by the sports community [8]. Esports has been considered a sport since 2003 in China, but in the USA, players also have official status as athletes [6].

### 1.1. Division and Characteristics of Games and Organization of Esports

All esports are video games, but not all games are esports. Esports refers to a classification of video games shaped around competition [9] involving amateur players, semi-professionals, and professionals, with substantial interest, income, and profit offered to support the best players [10].

Video games (especially action video games) can influence the cognitive domains of esports players [11]. Competitive video games also require the ability to engage in physical actions (e.g., clicking, typing, and button-mashing) within the virtual environment; these actions are called “mechanics” [12]. Many video games also contain theatrical and fictional elements, which are a part of the attraction. The game-playing experience of esports involves a unique combination of motor, cognitive, strategic, and mimetic skills [13]. The model of game competence is relevant for professional esports players because of certain features that are unique to the esports domain [6]. Currently, however, there is no scientific evidence available to create a model in this discipline. Esports can be divided into specific genres, each of which demand exceptional skills used in competitive games [14] that include multiplayer online battlefield arenas (MOBA), first-person shooters (FPS), real-time strategy games [6,15], and sports simulations [6]. Each genre involves different game characteristics, mechanics, competition policies [6], and relevant cognitive functions [16]. Examples of popular esports games include League of Legends, Counter Strike: Global Offensive, and Player Unknown’s Battlegrounds (PUBG). Competitive esports tournaments are organized for many people, mostly between professional players, and can take place in front of a live audience, via television broadcasting, or online [17]. During gameplay, the gamers must connect their perceptual–cognitive skills (e.g., anticipation, visual behavior, pattern recall, and action) and domain-specific skills (e.g., mouse and mouse movements) [15].

### 1.2. Health Risk and Dangers of Esports

Potentially adverse effects of esports games can arise from stressful, long-lasting training, as well as from the fierce competition that affects high-level players. According to Johnson and Woodcock (2021), world-class professionals are an integral part of the esports environment, but in the context of work, they are unstable, as they cannot maintain their highest skill levels after their thirties [10]. Interest in professional esports is especially prevalent among young people [18], as it is known that people over twenty years of age are not able to compete at the highest level because their reaction speed is slowed down [8]. Professional esports requires the playing of video games for several hours in order to improve skills such as gamepad and keyboard operation, game knowledge, strategy, and tactics [16]. Prolonged sitting may increase the risk of obesity and promote inappropriate eating behaviors [9]. Other esports requirements can also lead to serious health complications such as injury; eye strain; and neck, back, and wrist pain. Mental condition (depression and burnout symptoms) and elements of the psychosocial sphere (social anxiety, sleep disorders, and problems with personal hygiene) may also significantly deteriorate [16]. There are other games (e.g., exergaming) that, in their rules, require one to expend energy or physical activity, thereby improving the gamer’s physical condition and fitness. However, the level of energy expenditure is unknown among those participating in such games at a high level [18].

### 1.3. Potential Factors Affecting Success in Esports

Despite the ongoing development of the esports industry, there is a lack of scientific understanding regarding the determinants of high-level performance. Experts say there is a need for large-scale research to clearly determine the personal characteristics and factors that are important precursors to achieving satisfactory results in competition [3]. Recent studies in the field of esports psychology have identified the characteristics of successful players, including knowledge of the game, decision-making ability, motivation, the ability to separate private life from sports activity, concentration, emotional control, positive attitude, commitments to mental and physical warm-ups before training sessions, and flexibility in the relevant procedures and playing conditions. Moreover, to meet the requirements of esports, players improve their skills by training in various areas related to overcoming obstacles, teamwork, and/or individual physical effort, including the cognitive, psychological, physical, technical, and tactical domains [19]. Currently, the scientific literature indicates several cognitive domains such as attention (division and switching of attention), perception and processing of information (fast reaction time), and visual–spatial skills (navigation in a virtual environment) [16]. The selected physical characteristics (physical abilities) and health can also be considered important factors for success in sports [20].

Cognition is a key factor in athletic performance, and there is mounting evidence that select dietary ingredients, hydration levels, and supplements can affect brain function [21]. In the case of esports, the dominant role is played by cognitive processes, particularly executive functions, which help maintain long-term performance [22] and affect the achievement of satisfactory results [23]. To date, cognitive functions have been extensively studied for the prevention and support of treatments in neurodegenerative diseases [24,25,26,27,28,29,30,31], as well as to support healthy people [32,33,34,35,36,37]. Despite the high popularity of esports, there are still no publications in the field of physical culture sciences that explicitly and directly indicate the mechanism underlying the influence of nutrients and dietary supplements on the cognitive abilities and concentration of players. This review aims to characterize the relevant micronutrients and dietary supplements based on the available literature data to indicate what potential impact these micronutrients and dietary supplements may have on the cognitive functions of e-athletes. The first part of the article, published in 2020, describes the role of macronutrients and water in the context of cognitive functions in sport [38]. This review is a continuation of work in the field of micronutrients, i.e., vitamins, minerals, and biologically active substances.

## 2. Materials and Methods

A search was carried out using the following databases: PubMed and Medline, SPORTDiscus, Academic Search Ultimate, Scopus, and ScienceDirect for studies published in English from January 2011 to August 2021. The search terms contained keywords related to: “brain”, “nutrition”, “micronutrients”, “minerals”, “vitamins”, “supplements”, “e-sport”, “ESports”, “esports”, “esports science”, “cognition”, “cognitive functions”, “cognitive control”, “cognitive skills”, “video games”, “computer games”, and “physical activity”. The search resulted in 1727 articles that included randomized, double-blind, placebo-controlled trials, narrative review articles, systematic reviews, and meta-analyses. Then, the studies were selected based on the following criteria: studies published within 5 years (meta-analyses, systematic reviews, randomized control studies) and works examining cognitive functions. Based on the titles and abstracts, studies evaluating the use of dietary components in the prophylaxis of neurodegenerative or other neurological diseases, especially in the elderly, were excluded. Then, the databases were searched, taking into account dietary components such as: iodine, iron, zinc, magnesium, B vitamins, and vitamins C, D, and E, in addition to caffeine, L-theanine, polyphenols, beetroot juice, creatine, probiotics, lutein, and nootropic substances.

## 3. Results

### 3.1. The Influence of Dietary Micronutrients on Cognitive Functions

The diet provides trace elements, including iodine, iron, zinc, magnesium, B vitamins, and vitamins E and C, which play the role of cofactors in key enzymatic processes. They play an essential role in the metabolism of neurotransmitters, structural and functional brain lipids and proteins, DNA and RNA, and energy metabolism. Some micronutrients are necessary already at the stage of prenatal development for the proper development of the nervous system (iodine, iron, folic acid), which is related to the need to supplement the diet of pregnant women. Inadequate intake of micronutrients in adulthood leads to deficiencies, and in case of long-term insufficient supply of a given ingredient, it may lead to the development of many diseases or deterioration of well-being. Given that certain deficiencies, such as iron and iodine, are fairly common even in developed countries [39,40], ensuring adequate micronutrient intake is an essential step in promoting optimal cognitive functioning.

#### 3.1.1. Minerals

##### Iodine

Iodine is critical to the developing brain as a component of the thyroid hormones thyroxine (T4) and triiodothyronine (T3) [41,42]. The importance of adequate consumption of this micronutrient is especially important during neurological development [43,44,45,46]. Iodine deficiency can cause a decrease in brain mass and a greater cell mass in the cortex, and a lower cell mass in the baby’s cerebellum. Neurological cretinism mainly begins with maternal hypothyroidism due to iodine deficiency. Even moderate or mild hypothyroxinemia during pregnancy increases the risk of neurodevelopmental deficits in the offspring [47]. Sufficient iodine intake is needed for the synthesis of adequate amounts of thyroid hormones. There is evidence to suggest that low levels of T4 may adversely affect the development of the fetal brain and, subsequently, the cognitive functions of children and adults [48].

Santiago-Fernandez et al. conducted a study to assess the intelligence quotient (IQ) of students aged 6 to 16 living in a province with mild to moderate iodine deficiency in Spain. The children were divided according to the concentration of iodine in the urine. The obtained results showed that children with urine iodine concentration <100 μg/L had a lower IQ compared with children with urine iodine concentration above 100 μg/L [49] (Table 1).

Two randomized, double-blind, placebo-controlled studies have shown an association between iodine deficiency and cognition in school-age children. Zimmermann et al. conducted a study among children from 10 to 12 years of age from Southeast Albania with moderate iodine deficiency. Initially, the iodine concentration in the urine of the children was evaluated and seven cognitive tests were performed. Children were provided with iodized poppy seed oil (containing 400 mg of potassium iodate) or a placebo (sunflower oil). After six months, it was observed that iodine nutritional status in the iodine supplementation group improved from moderate deficiency to adequate concentration, while the control group was moderately deficient in iodine throughout the study period. The experimental group experienced significant improvement in four out of seven cognitive tests compared with the placebo group [50] (Table 1).

A similar study was conducted by Gordon et al. in children aged 10 to 13 with mild iodine deficiency living in New Zealand. The children were randomly split into two groups: supplementing them daily for 28 weeks, iodine in the form of tablets containing 150 μg/L of iodine, and the control group receiving a placebo. Initially, cognitive function was evaluated using the four subtests of the Wechsler Intelligence Scale for Children. After the end of the intervention, members of the iodine supplementation group had an increase in iodine concentration to a sufficient amount, while participants in the control group were still moderately deficient in iodine. A significant therapeutic effect was found in two out of four cognitive tests in the iodine supplement group compared with the placebo group, which suggests that iodine deficiency affects the development of cognitive functions until at least 13 years of age [51] (Table 1).

Apart from studies on pregnant women and the effects of iodine nutrition on fetal development and cognitive function of the newborn, there have been few studies involving adults. In adults, thyroid hormones affect mood, may induce affective disorders and dementia, and could potentially be associated with cognitive functions including memory and attention [62]. Both in healthier, older adults with regular thyroid function and in older adults with hypothyroidism, a positive relationship has been found between the level of thyroid hormones and the results of cognitive functions [63,64].

The best sources of iodine in your diet are sea fish and seafood, algae, and dairy products, including milk, cheese, yogurt, and eggs. In connection with people’s personal food choices, iodization of table salt is a useful method for preventing deficiencies in this nutrient [65,66,67].

##### Iron

Iron is a trace mineral that participates in many physiological processes, such as oxygen transport, oxidative metabolism, energy production, cell growth and differentiation, DNA synthesis, cellular immune response, and many others [68]. It is also a key mineral that is essential for the differentiation and proliferation of neurons. Iron deficiency affects neuronal processes such as myelination, dendritic branching/branching, and neuronal plasticity. Moreover, iron deficiency influences iron allocation in the brain, thus altering the prefrontal–subcortical dopaminergic and frontal striatal networks that mediate cognitive functions [69]. Disturbances in dopaminergic systems, therein the nigrostriatal and mesocortical pathways, may contribute to the formation of cognitive deficits, in particular difficulties in visual–spatial processing related to iron deficiency [52] (Table 1). This was confirmed in a study by the Carpenter team. Researchers found a positive relationship between spatial IQ and the average amount of iron in the basal ganglia, and in particular in the caudate, suggesting that the presence of iron in certain regions of iron-rich deep basal ganglion nuclei influences spatial intelligence [53] (Table 1).

Iron deficiency in the neonatal period and infancy is considered a factor predisposing to the development of cognitive development disorders. According to the results of experimental studies, iron deficiency may cause cognitive deterioration in animals and humans [70,71], with damage to the brain’s mitochondria as the basis for these changes [72]. In addition, there are presumed changes in dopamine metabolism in the brain, as well as altered serotonergic neurotransmission and changes in dopamine receptors. In fact, iron deficiency or anemia causes severe changes in dopamine synthesis in key areas of the brain [73]. The cognitive disorders caused by iron deficiency include mainly those related to concentration of attention, intelligence, and sensory perception function, as well as those related to emotions associated with iron deficiency anemia. Moreover, iron deficiency beyond anemia could possibly cause cognitive impairment [54] (Table 1).

A meta-analysis of 32 studies involving 7089 children aged 5–12 years by Low et al. showed that iron supplementation improved general cognitive functions and IQ in addition to attention and concentration measures in children with anemia [55] (Table 1). In another study in 140 Indian teens, intake of iron-biofortified pearl millet resulted in greater improvements in attention and memory compared with the group consuming regular pearl millet. Reaction time on attention-assessment tasks was doubled in adolescents consuming biofortified millet for 6 months compared with conventional millet [56] (Table 1). In their systematic review, Lomango and co-authors analyzed 10 randomized controlled trials and 1 non-randomized control trial in premenopausal women between 12 and 55 years of age. Among the analyzed studies, as many as seven showed an improvement in mood and cognitive functions after iron supplementation [57] (Table 1). Conversely, a study of 428 Chinese 12-year-olds showed that both iron deficiency and high iron levels contributed to the reduction in neurocognitive performance in a discipline-specific way in early adolescents. Iron deficiency compared with the normal group was associated with slower performance of tasks that measured abstraction, mental flexibility, and spatial processing capacity. High serum iron concentration was associated with lower accuracy in the task of measuring spatial processing abilities and a longer reaction time in the task evaluating abstraction and mental flexibility compared with normal serum levels [58] (Table 1). These results show that it is important to maintain an adequate level of iron in the serum for the proper functioning of the brain. Disruption of the iron homeostasis may result in its accumulation in the brain and interaction that enhances oxidative damage [59] (Table 1). Cognitive dysfunction associated with high iron levels is likely due to iron cytotoxicity to brain function [74].

The source of heme iron is red meat, poultry, liver, offal, fish, and egg yolk. Non-heme iron comes from plant products—vegetables, legumes, and whole grains, especially millet and amaranth, pumpkin seeds, and cocoa and cocoa products [67].

##### Zinc

Zinc is an essential trace element that plays a vital role in brain function. Free zinc ion (Zn^2+^) neurons are found in various areas of the brain, including the cerebral cortex, amygdala, olfactory bulb, and hippocampal neurons. Zinc plays a key role in enzymatic activity, cell signaling, and modulation of neurotransmitter activity [75]. It is considered essential for the formation and migration of neurons and for the formation of neuronal synapses [76]. Adult brain neurogenesis is dependent on the presence of zinc, which has wide-ranging implications for hippocampal function, including learning and memory, as well as emotional and mood control [75]. One of the few studies available on the relationship of zinc and cognitive performance in adults examined zinc supplementation (15 or 30 mg/day) in 387 healthy adults aged 55–87 years and found that each dose of zinc used for 3 months had a beneficial effect on spatial working memory. However, the benefits of supplementation were not observed in the case of other examined indicators [32] (Table 1). Warthon-Medina et al. conducted a meta-analysis of 18 studies that showed no significant effect of zinc supplementation on cognition in children, although there were some modest signs of improvement in aspects of executive function and motor development after the supplementation began [60] (Table 1).

The products rich in zinc include oysters, beef, legumes (e.g., beans, chickpeas), yoghurts, ripening cheeses, and cereals [67].

##### Magnesium

Magnesium is a mineral vital to the proper functioning of all human cells, including neurons. It takes part, among others, in many enzymatic reactions [77], intracellular transmission [78], myelination [79], and synaptic formation and maintenance [80] and in the regulation of serotonergic, dopaminergic, and cholinergic transmission [81]. In the nervous system, magnesium is therefore essential for neuromuscular coordination and optimal nerve transmission. It plays a role in protecting against overstimulation that leads to cellular death: It interacts with the aspartate receptor to block the calcium channel at that receptor and must be cleared for glutamatergic excitatory signaling to occur. Low levels of magnesium may theoretically increase glutamatergic neurotransmission, which may lead to oxidative stress and nerve cell death [82]. The vast majority of studies conducted so far have focused on depression, due to the recognized role of magnesium in several basic mechanisms of the pathophysiology of depression, including inflammation and oxidative stress [61] (Table 1). The majority of preliminary symptoms of a magnesium deficiency include neurological or neuromuscular symptoms. Neuromuscular hyperactivity, including muscle cramps, is a prevalent characteristic of magnesium deficiency, but latent tetany, generalized seizures, dizziness, and muscle weakness may also occur [83]. Other symptoms of magnesium deficiency can manifest as tiredness, lethargy, lightheadedness, and appetite deficit [84].

The products are obtained from magnesium include: hard water, bananas, soybeans, nuts, spices, green leafy vegetables, apricots, and whole grains [67].

#### 3.1.2. Vitamins

##### B Vitamins

The group of B vitamins includes eight water-soluble vitamins that are vital in the context of cell function, acting as coenzymes in many enzymatic reactions. Their joint action is well known in many aspects of brain function—that is, energy production, DNA and RNA synthesis and repair, genomic and non-genomic methylation, and the synthesis of diverse neurotransmitters and signaling molecules [85]. The body has no other organ so active metabolically than the brain. Though it accounts for just 2% of body weight, it is responsible for more than 20% of total energy consumption [86]. Therefore, the metabolic functions of B vitamins and their role in neurochemical synthesis can be viewed as having a special impact on the functioning of the brain [85].

Thiamin (vitamin B1) is a part of synaptic formation, axonal growth, and myelin genesis, leading to the formation of functional neuroglia. It also can maintain the membrane of newly formed neurons during embryogenesis and can control apoptosis [87]. It is essential for the synthesis of fatty acids, steroids, nucleic acids, and aromatic amino acids, which are the precursors of a number of neurotransmitters, therein acetylcholine, glutamate, and gamma-aminobutyric acid [88]. Food sources of thiamine are liver, pork, eggs, nuts, oats, oranges, dry pulses, yeast, and powdered milk [67]. Folic acid (vitamin B9) is important in the maintenance of the lipids of the neuronal and glial membranes, which can affect brain function. This may manifest as changes in mood, sleep rhythm, and irritability [89,90]. Food sources of folic acid are avocado, dark vegetable leaves, spinach, pawpaw, oranges, seeds, and nuts [67]. Pyridoxine (vitamin B6), folic acid (vitamin B9), and cobalamin (vitamin B12) are essential cofactors for the synthesis of myelin and neurotransmitters [91]. Food sources of pyridoxine are potatoes, fruit, fish, beef liver, and other parts of meat [67].

Deficiency of folic acid, mainly caused by low intake, is associated with a number of physiological irregularities during development and adulthood. Proper folate levels are essential for brain function, and insufficient folate supply can lead to neurological disorders such as depression and cognitive decline. Folate supplementation, alone or in combination with other B vitamins, has proven effective in prevention of cognitive decline and dementia during aging and in enhancing the effects of antidepressants. The results of a randomized clinical trial showed that a 3-year folic acid supplementation can positively influence the reduction in aging-related cognitive decline [92] (Table 2).

Nutritional deficiencies of micronutrients such as vitamin B12, folate, and zinc may cause symptoms of depression and dementia, including depressed mood, fatigue, cognitive decline, and irritability [106,107]. Vitamin B12, vitamin B6, and folic acid play an important role in the metabolism of plasma homocysteine (tHcy), and vitamin deficiency hyperhomocysteinemia is associated with cognitive decline later in life. Cross-sectional, case–control, and long-term observational studies have indicated that high tHcy raises the risk of cognitive impairment with or without dementia, but the nature of these studies makes it challenging to conclude whether these associations are actually causal. Increased tHcy is associated with an increased risk of cognitive impairment and dementia, although the available evidence from RCTs does not indicate any obvious cognitive benefit from lowering tHcy with B vitamins [93] (Table 2).

A study of 3136 American adults aged 18 to 30 was conducted. Study participants were followed for 20 years. People with the highest dietary niacin intake at baseline (i.e., median intake 24.7 mg/day) scored better in cognitive tests 20–25 years later than those with the lowest dietary niacin intake (median 8.6 mg/day, so below the US recommendations, set at 15 mg per day). The study participants with the highest dietary intake of pyridoxine at baseline (i.e., mean intake of 3.0 mg/day) performed better in psychomotor speed tests 20–25 years later compared with those with the lowest dietary pyridoxine intake (median 0.7 mg/day). Young adults consuming large amounts of dietary folate (mean consumption: 384 g/day) showed better cognitive functions 20–25 years later in middle age, compared with those with the lowest dietary folate intake (mean consumption: 152 g/day). Those with the highest dietary intake of vitamin B12 at baseline (i.e., mean intake of 8.7 g/day) surpassed those with the lowest dietary cobalamin intake (median 2.2 g/day) in psychomotor speed 20 to 25 years later [94] (Table 2).

In a group of 317 healthy Korean children without prior diagnosis of neurological or psychiatric disorders, analyses suggested that vitamin B1, B2, B6, and niacin intake was negatively correlated with omission errors that indicated inattention [95] (Table 2). Louwman et al. conducted a study to assess whether cognitive functions in adolescents (10–16 years old) with marginal cobalamin status were impaired because of a macrobiotic diet up to 6 years of age. Control subjects who followed a standard diet since birth, without elimination of meat products, achieved better results on most psychological tests than macrobiotic subjects with low or normal cobalamin status [96] (Table 2). Cross-sectional studies in older children or adolescents most often showed that lower levels of biochemical vitamin B12 were correlated with deterioration of academic performance and mental and social development, as well as poorer short-term memory and attention [97] (Table 2). A systematic review published in 2021, focusing on vitamin B12 supplementation in elderly patients with regular or subclinical serum levels of vitamin B12 and without advanced neurological disorders, found no effect of treatment on cognitive functioning and symptoms of depression [98] (Table 2).

##### Vitamin E

Vitamin E (α- and γ-tocopherols) is an essential fat-soluble vitamin and antioxidant. It inhibits the propagation of the lipid oxidation chain reaction, which may particularly affect polyunsaturated fatty acids in the cell membrane, along with neuronal structures [108]. It affects cognitive performance, and a decrease in serum vitamin E levels was associated with poor memory performance in the elderly [109]. A meta-analysis of randomized controlled trials did not show a significant effect of vitamin E on cognitive functions in middle-aged and elderly people without dementia [99] (Table 2). A meta-analysis of randomized controlled trials investigating oral vitamin or mineral supplements in participants diagnosed with mild cognitive impairment showed that three years of treatment with high doses of vitamin E probably did not reduce the risk of progression to dementia. However, only one study was included in the analysis, which analyzed the effect of vitamin E supplementation on cognitive functions, which requires confirmation in further studies in the future [100] (Table 2). The sources of vitamin E are nuts (almonds, hazelnuts, peanuts), vegetable oils, soybean oil, wheat germ, safflower, sunflower seeds, spinach, and broccoli [67].

##### Vitamin D

Vitamin D plays a key role in the metabolism of bone and calcium in the human body. It also serves other functions in the body, including modulation of cell development, neurogenesis, neuroprotection, detoxification, immune function, and reduction in inflammation [110]. The vitamin D receptor located in the brain is involved in the complex planning, processing, and formation of new memories [111], which would herald the importance of cognitive functions and a better overall neurological state [112]. The occurrence of vitamin D deficiency is a major global public health challenge. Based on their research, scientists have linked vitamin D to disorders affecting multiple body parts, such as cardiovascular diseases, cancer, stroke, and metabolic disorders, not to mention diabetes [113]. In a randomized trial conducted, Mendel Maddock et al. showed that there is no evidence that serum [25(OH)D] concentration is a factor influencing cognitive functions in middle and later life [101] (Table 2). Concentrations of [25(OH)D] below 25 nmol/L increase the risk of dementia, especially in adults and patients over 65 years of age [102] (Table 2). According to a meta-analysis by Goodwill et al., low vitamin D levels are associated with poorer cognitive functions; however, interventional studies have not shown clear benefits of vitamin D supplementation [103] (Table 2). In a study involving younger, healthy adults, 128 people were divided into two groups. The first group (63 people) took 5000 IU/day of cholecalciferol for 6 weeks, while the second group (65 people) took placebo for 6 weeks. The aim of the study was to assess the effect of supplementation on cognitive functioning and secondary emotional measures. Subjects were assessed for working memory, response inhibition, and cognitive flexibility at baseline and at 6 weeks. As a result of the intervention, the group receiving vitamin D significantly developed serum levels of [25(OH)D], while no difference was observed in the group receiving placebo. No measure of cognitive functioning improved after 6 weeks of follow-up [104] (Table 2).

In another clinical study in healthy adults, 82 patients with a baseline value of [25(OH)D] ≤100 nmol/L were randomized to one of two groups. The first group received a high dose (4000 IU/day; *n* = 42) of vitamin D3 for 18 weeks, and the second group received a low dose (400 ID/day; *n* = 40) to assess the effects of supplementation on cognitive functions according to the Symbol Digit Modalities test, assessing verbal fluency and digit span, and the Cambridge Automated Neuropsychological Test Battery (CANTAB) battery. Serum [25(OH)D] levels increased significantly more in the high-dose group. Improvement in non-verbal (visual–spatial) memory performance in the high-dose group and in the group with lower baseline [25-(OH)-D] levels (<75 nmol/L) improved significantly. The results suggest that elevated [25(OH)D] levels are crucial for executive functions such as non-verbal memory [35] (Table 2). Sources of vitamin D in food are fortified milk, cheese, cereals, egg yolk, salmon, and fortified margarine [67].

##### Vitamin C

Vitamin C participates in the synthesis and modulation of various hormonal components of the nervous system. It is a catalyst for enzymes that catalyze the formation of catecholamines (noradrenaline and adrenaline) and enzymes active in the biosynthesis of neuropeptides [114]. A group of components of the nervous system are adjusted by ascorbate (vitamin C) levels, therein neurotransmitter receptors and brain cell structures (such as glutamatergic and dopaminergic neurons) and the synthesis of glial cells and myelin [114,115]. Travica et al. reviewed a systematic review of 50 randomized controlled trials to find the relationship between vitamin C status and cognitive efficiency in both healthy (cognitively intact) and disabled individuals. Of these, 36 studies were conducted in healthy participants and 14 in people with cognitive impairment (including Alzheimer’s disease and dementia). Vitamin C status was measured using meal frequency questionnaires or direct assessment of vitamin C in plasma. Cognitive function was studied using various tests, mainly the Mini-Mental-State-Examination (MMSE). The results of the study showed higher mean vitamin C concentrations in the groups of cognitively intact participants compared with the groups with cognitive impairment. There was no correlation between vitamin C concentration and the cognitive function of MMSE in people with cognitive impairment. MMSE was not adequate to detect cognitive variance in the healthy group. An analysis of studies that used different cognitive assessments in cognitively intact subjects was not the subject of this review; however, a qualitative assessment showed a potential relationship between vitamin C levels in plasma and cognitive function [105] (Table 2). Sources of vitamin C are citrus fruits (lemons, oranges, limes), papaya, red and green peppers, tomatoes, kiwi, strawberries, cantaloupes, leafy vegetables and their juices, fortified cereals, and potatoes [67].

### 3.2. The Influence of Dietary Supplements on Cognitive Functions

There is growing evidence that certain dietary supplements may positively influence cognitive performance. These include caffeine, L-theanine, creatine, polyphenols, beetroot juice, probiotics, lutein, nootropics, and other plant supplements.

#### 3.2.1. Caffeine

Caffeine is a purine alkaloid and a very popular psychostimulant that is widely used by people all over the world. It is found in coffee beans, cola nuts, tea leaves, and yerba mate, as well as guarana seeds and cocoa beans [116]. It has been classified as a supplement with a very well documented effect, used to improve exercise performance. The benefits of its supplementation are observed in terms of sports performance in situations requiring endurance and in short-term, supra-maximum, or repeated sprint tasks [117]. Caffeine influences cognitive and physical functions by blocking adenosine A1 and A2a receptors in the central nervous system and peripheral tissues. Doses of 1–4 mg/kg body weight improve alertness, concentration, and reaction time [118] (Table 3), while doses of 3–6 mg/kg b.w. caffeine can enhance cognitive performance, motor skills, and physical performance in many types of sports [117] (Table 3).

The effect of caffeine on cognitive performance is also being investigated. This term covers executive functioning (EF), decision making, and creativity. Zhang et al. investigated the effects of low, moderate, and high doses of caffeine on cognitive performance and brain activation. Low dose caffeine intake (3 mg/kg b.w.) had a greater effect on cognition and brain activation than moderate and high doses of caffeine (6 and 9 mg/kg b.w.), suggesting low-dose caffeine can be a discriminating supplement in enhancing function executive and prefrontal activity [119] (Table 3).

Lorenzo Calvo et al. investigated the relationship between caffeine consumption and cognitive performance during sports. The systematic review included 13 studies assessing the effect of caffeine on objective measures of cognitive performance or cognitive function. Five of these studies were also meta-analyzed. After collecting the data in a meta-analysis, a significant effect of caffeine was only revealed in terms of attention, accuracy, and speed. However, the results of 13 studies suggest that consuming a low to moderate dose of caffeine prior to and/or during exercise may enhance mood and cognitive functions such as attention; it may also increase simple reaction time, choice reaction time, memory, or fatigue, but this is dependent on research protocols [120] (Table 3).

#### 3.2.2. L-Theanine

Tea, a drink made from the leaves of *Camellia sinensis*, has enjoyed wide consumption throughout human history. Many studies show that tea intake is associated with beneficial effects on brain health, including a reduced incidence of cognitive deterioration and decreased incidence of depression and mental stress [154,155].

L-Theanine is a non-protein amino acid found naturally in tea and has been mainly studied for its effects on the brain. After crossing the blood–brain barrier, L-theanine influences the central nervous system (CNS) by affecting neurotransmitters and attenuating stress-related CNS responses [156]. Haskell et al. investigated that a combination of 250 mg of L-theanine and 150 mg of caffeine improved reaction time, working memory, and accuracy of task verification in a group of volunteers aged 18–34 years [121] (Table 3). The meta-analysis by Camfield et al. collected 11 human studies that investigated the effects of L-theanine alone or in combination with caffeine on cognitive ability and mood. Caffeine in combination with L-theanine had a beneficial effect on the concentration and emotional state of the study participants [122] (Table 3). Dietz and Dekker reviewed the available studies, concluding that L-theanine and caffeine had beneficial effects on continuous attention, memory, and distraction suppression. In contrast, L-theanine leads to relaxation by reducing caffeine-induced arousal [123] (Table 3). The cognitive effects of green tea are related to the combined effects of caffeine and L-theanine, while separate administration of each substance has been shown to have a smaller effect [124] (Table 3). In another study, Zaragoza et al. attempted to test the effect of supplementation with a low-dose combination of caffeine, theanine, and tyrosine on sport-specific cognition in tests where accuracy of movement and reaction time are important. The results obtained in the study showed that the combination of low-dose caffeine with theanine and tyrosine can improve athletes’ accuracy of movement and reaction time during a series of strenuous exercises [125] (Table 3).

#### 3.2.3. Polyphenols

Polyphenols are secondary metabolites of plants and can be divided into four categories: phenolic acids, flavonoids, lignans, and stilbenes. They play a protective role in neurodegeneration and interact with neuronal signaling pathways [157,158]. Fruits and drinks such as tea, red wine, cocoa, and coffee are the main sources of dietary polyphenols. They affect peripheral and cerebrovascular blood flow, interact with intracellular neuronal and glial signaling, and help protect neurons from damage [159,160,161]. Research results show that polyphenols have a positive effect on cognition and memory and reduce neuronal damage [162,163,164].

Polyphenols such as flavonoids and resveratrol have been shown to exert many important effects on the brain, including modulation of cellular pathways involved in gene expression, neuroprotection, neuroplasticity, and endogenous antioxidant defense and increasing cerebral blood flow [165,166]. The intake of polyphenols with the diet is associated with a reduced risk of developing dementia, improving cognitive abilities in the aging process [167]. The mechanisms of action of polyphenols include the reduction in oxidative stress and inflammation and the improvement in cerebral perfusion [126,127] (Table 3). Later studies suggested that berry polyphenols may have multiple effects in addition to their anti-inflammatory and antioxidant functions [168]. Additionally, it has been shown that anthocyanins contained in blueberries penetrate the brain, and their concentration has been correlated with cognitive performance [169].

Hepsomali et al. conducted a systematic review and meta-analysis to investigate whether polyphenol consumption might have a beneficial effect on cognition, especially on accuracy and speed of attention. A total of 18 placebo-controlled human intervention studies were included in the meta-analysis. The results of the study indicate that acute polyphenol consumption may improve the speed of processing visual information, a higher-order task with vigilance elements, working memory, and executive functions in young participants [128] (Table 3).

#### 3.2.4. Cocoa Flavanols

Cocoa beans are commonly known for being a primary source of phenolic compounds with unmatched content of flavanols of any food on a per weight basis [170]. More and more studies confirm the beneficial effects of cocoa and cocoa-containing products on human cognition. The cognitive benefits of consuming chocolate are identified in a systematic review that reported improvements in cognitive or task performance among young adults (under the age of 25) and children who regularly consumed chocolate. This effect is attributed to polyphenols, including flavanols [129] (Table 3). Decroix showed in a conducted study involving 12 healthy men that the beneficial effect of cocoa flavanols on brain oxygenation at rest was offset by a strong increase in brain perfusion and oxygenation induced by exercise. Cocoa-derived flavanols had no additional effect on exercise-induced cognitive enhancement and the associated increased brain oxygenation and perfusion [126]. A study in 98 healthy adults found that consuming 35 g of dark chocolate with a cocoa content of 70% may have a positive effect on episodic verbal memory compared with an equal-calorie bar of white chocolate of 35 g two hours after ingestion in healthy young adults [171]. According to a review by Socci et al., the ad hoc supply of cocoa flavanols may result in an immediate improvement in cognitive functions, as well as maintaining the efficiency of cognitive functions under situations of fatigue and insufficient sleep. [130] (Table 3). Tan et al. conducted a systematic review of 18 studies to assess the health effects of cocoa and cocoa-containing products. The consumption of chocolate or cocoa products significantly improved lipid (triglyceride) profiles, while the effects of chocolate in all other outcome parameters did not differ significantly, including no significant effect on cognitive functions [131] (Table 3).

#### 3.2.5. Beetroot Juice

Beetroot juice is a scientifically proven supplement. It is a popular additive that has been widely studied for its effects on improving the ability to perform long-term submaximal training [172] and intense, intermittent, short-term exercise [132,173] (Table 3). Its beneficial effect in the context of sports is due to the high content of nitrates (NO_3_^−^), compounds found naturally in vegetables. Dietary nitrates are reduced endogenously to nitric oxide. Most studies suggest that beetroot juice supplementation is vital in modulating skeletal muscle function [174], increases exercise capacity through increased function of type II muscle fibers [175], reduces ATP cost of muscle strength production, increases mitochondrial respiration efficiency, and increases blood flow through blood vessels [176].

One study found that dietary nitrates increase performance during repeated sprints and may ameliorate the cognitive decline (particularly in reaction time) that may occur with prolonged repetitive exercise [132]. In another randomized, double-blind, placebo-controlled, parallel group study, 40 healthy adults received a placebo or 450 mL of beetroot juice. The results of the study show that single doses of dietary nitrate can modulate the brain’s blood flow response to task performance and potentially improve cognitive performance and suggest one possible mechanism by which vegetable consumption may have a beneficial effect on brain function [133] (Table 3). Stanaway et al. conducted a study to assess whether supplementation with nitrate-rich beetroot juice would improve cardiovascular and cognitive functions in elderly people (50–70 years old) and young adults (18–30 years old). The results indicate that beetroot juice supplementation can lower blood pressure and boost some aspects of cognitive performance, thus providing potential health benefits for both age groups [134] (Table 3).

#### 3.2.6. Creatine

Creatine is a supplement used by athletes to improve efficiency in sports by increasing the energy supply of muscle tissues. It is also a crucial component of the brain, and according to research, it supports cognitive processes, enhancing energy supply and influencing neuroprotection [135]. Brain cells use adenosine triphosphate (ATP) to produce energy. The phosphocreatine and creatine (PCr/Cr) system is linked to adenine nucleotides through the creatine phosphokinase reaction [135] (Table 3). Overall, creatine is involved in cellular energy homeostasis [177,178]. The results of many studies have shown that creatine supplementation increases the PCr content in the brain by 5–15%, and thus improves the brain’s bioenergetics [179]. Watanabe et al. found that dietary supplementation with creatine (8 g/day for 5 days) reduces mental fatigue when subjects repeatedly perform simple mathematical calculations [136]. Rea et al. investigated whether oral creatine supplementation (5 g per day for six weeks) increased IQ scores and working memory performance in 45 young adult vegetarians. Creatine supplementation had a significant positive effect on both working memory and intelligence. Both tasks required processing speed [137] (Table 3). McMorris et al. found that creatine supplementation (20 g/day for 7 days) had a positive effect on the mood state and performance of tasks that affect the prefrontal cortex in sleep-deprived participants [138] (Table 3). Ling et al. investigated whether creatine ethyl ester supplementation (5 g/day for 15 days) might improve cognitive functions in some tasks [139] (Table 3). Creatine supplementation may improve the performance of tasks that require memory and intelligence [135] (Table 3). While the role of creatine supplementation in brain function is promising, more research is needed, as evidence suggests that the blood–brain barrier is an obstacle to the circulation of creatine, which may require higher doses and/or longer protocols to increase the concentration of creatine in the brain in comparison with the muscles. Not all research results are unequivocal. Benton et al. showed that creatine supplementation did not affect the indicators of verbal fluency and alertness. However, in vegetarians, supplementation with creatine resulted in better memory compared with the group consuming meat [140] (Table 3).

Studies show that guanidinoacetic acid (GAA, 3 g/day), a naturally occurring creatine precursor, shows superior results in increasing brain creatine content compared with an equimolar dose of creatine [141] (Table 3), although it should be noted that this finding was based on a study involving five participants.

Creatine may be used in stressful situations when a temporary decrease in creatine levels may occur to potentially offset the negative cognitive effects of sleep deprivation [142] (Table 3). The effect of creatine supplementation on brain function seems to be greater under stressful conditions that lead to temporary (e.g., mental fatigue, exhausting exercise) or chronic (e.g., aging, depression) creatine depletion, while the effect is absent or minimal in healthy people under non-stressful conditions [143]. A review by Roschel et al. concluded that supplementation with creatine may enhance cognitive processing, especially in circumstances of creatine deficits in the brain that may be caused by acute stressors (e.g., exercise, sleep deprivation) or chronic pathological conditions (e.g., deficiencies of creatine synthesis enzymes, mild traumatic brain injury, aging, Alzheimer’s disease, depression). Nevertheless, the optimal creatine protocol capable of increasing brain creatine levels remains to be determined [144] (Table 3).

#### 3.2.7. Probiotics and Prebiotics

There is a clear relationship between probiotics, psychobiotics, and cognitive and behavioral processes that include neurological, metabolic, endocrine, and immune signaling pathways. Changes in these systems can cause changes in behavior (mental disposition) and cognitive level (learning and memory) [180]. The microbiota–gut–brain axis is the interaction between the gut microbiota and the brain. This combination has a great impact on human health, including on mental illness and on behaviors that affect mental health and cognitive performance [181,182,183].

Intestinal colonization begins at birth and continues through the first 3 years of life. The initial interaction between the gut microbiota and the host is essential for the maturation of the nervous system, the immune system, and development regulation of intestinal physiology [184,185,186]. At this stage, the intestinal microbiota is also able to modulate the angiogenesis process [187]. In addition, the microorganisms also exhibit antimicrobial activity, thus maintaining a stable intestinal ecosystem. It has been shown that changes in the microbial colonization of human intestines early in life influence the risk of developing diseases [188]. Later in life, microbial colonization of the gut has significant effects on the host’s neurophysiology, behavior, and nervous system function [181,189,190]. The gut microbiota, thanks to its immunomodulatory properties, influences brain function and human behavior through various immune pathways, inside and outside the central nervous system (CNS). It has been shown that microbial neuroimmune modulation may contribute to etiopathogenesis or show important signs and symptoms in neurodegenerative and behavioral disorders such as autism spectrum disorders, anxiety, depression, Alzheimer’s disease, and Parkinson’s disease [189]. Probiotic interventions provide specific strains of bacteria that can influence composition and activity of the gut microflora. In a study of 45 healthy adults, a 4-week administration of a multi-strain probiotic intervention (7.5 × 106 CFU/g) modulated functional activity in areas of the brain associated with higher-order cognitive processes such as problem-solving, reasoning, attention, decision-making, learning, and creativity [145] (Table 3).

Prebiotics, as defined by the International Scientific Association for Probiotics and Prebiotics, are “a substrate that is selectively used by host microorganisms to provide health benefits” [191]. Dietary ingredients with prebiotic properties provide substrates for the metabolism of commensal bacteria, unlike probiotics, which provide external strains of probiotic bacteria. The prebiotic preparations include, among others carbohydrate-based compounds such as fructooligosaccharides and inulin, as well as polyphenols and polyunsaturated fatty acids [192]. An additional intervention aimed at providing probiotics and prebiotics, as well as bioactive metabolites produced during fermentation processes, is the use of fermented products such as yogurt, kefir, and sauerkraut. Certain probiotic bacteria found in fermented foods can initiate the production of the neurotransmitter gamma-aminobutyric acid [193].

A systematic review of 14 prebiotic intervention studies aimed to evaluate the available prebiotic intervention studies in humans on cognition and affective functions, highlighting the potential mediating role of microbiota. Some chronic prebiotic interventions (>28 days) improved affect and episodic verbal memory compared with placebo. Acute prebiotic interventions (<24 h) were more effective in improving cognitive variables (e.g., verbal episodic memory). [146] (Table 3). A meta-analysis by Marx et al. aimed to evaluate randomized control trials that investigated the use of probiotic, prebiotic, and fermented food interventions to improve cognitive function. The results of the analysis do not show a benefit of using probiotic, prebiotic, and fermented food interventions for cognitive outcomes. However, this could be due to the great diversity of the population, types of cognitive tests, and introduced interventions; therefore, there is a need for further research [147] (Table 3). A randomized control study in 18 women was designed to test the effectiveness of dietary fiber, polydextrose, in improving cognitive performance and acute stress responses by manipulating the gut microflora in a healthy population. Supplementation with polydextrose caused a slight improvement in cognitive performance. The results indicate that polydextrose may positively influence communication between the gut and the brain and may modulate behavioral responses [148] (Table 3).

Exposure to chronic psychosocial stress reduces the level of *Bacteroides* spp. and increases the level of *Clostridium* spp. in the caecum, while increasing the level of circulating interleukin IL-6 and CCL2 (monocyte chemotactic protein, MCP-1), indicating immune activation. Levels of IL-6 and CCL2 correlate with changes in the levels of *Coprococcus* spp., *Pseudobutyrivibrio* spp., and *Dorea* spp. induced by stressors directly in the intestine [194]. The aim of the probiotic intervention undertaken by Bloemendaal et al. was to evaluate how probiotics can buffer the detrimental effects of stress on cognition by studying the association with probiotic-induced changes in the gut microbiota. Intestinal microbiological changes after 28-day supplementation with multi-strain probiotics (ecologic barrier consisting of *Lactobacilli*, *Lactococci*, and *Bifidobacteria* in healthy women (probiotic group *n* = 27, placebo group *n* = 29) were analyzed. People with a higher increase in the number of Ruminococcaceae_UCG-003 after probiotics were better protected against the negative effects of stress on working memory after probiotic supplementation [149] (Table 3).

#### 3.2.8. Lutein

Lutein and its analogue zeaxanthin belong to the group of carotenoids that occur naturally in food. Rich sources of lutein include einkorn, Khorasan, and durum wheat and maize, as well as the food products that contain these substances. These carotenoids are also found in green leafy vegetables and egg yolks [195]. Lutein is responsible for processes related to vision, as well as for protecting the eyes against the development of diseases. Current evidence suggests that lutein may have a beneficial effect on the optimization of vision and cognitive functions at every stage of life [150,151] (Table 3). It is presumed that the substance’s antioxidant and anti-inflammatory effects are responsible for this activity [151] (Table 3).

#### 3.2.9. Other Plant Supplements

There is a lot of research on the influence of phytonutrients on the work of the brain and on cognitive functions. There are many plant products, herbal extracts, or phytonutrients such as ginseng, mint, *Ginkgo biloba*, *Bacopa monnieri*, lion’s mane, *Rhodiola rosea*, guarana, rosemary, saffron, turmeric, ashwagandha, and xanthines, which are sold as tonic supplements to enhance cognitive functions. Many of the dietary ingredients listed appear to have promising effects on cognition. Several studies have shown positive effects of these products [1,21], but more research is needed.

Nootropics (“smart drugs”) have a documented effect on cognition. Their mechanism of action strengthens mental functions such as memory, creativity, motivation, attention [196], concentration, and speed [197], which are important from the perspective of esports. The competitive nature of esports has led players to choose various nootropics or drugs that improve cognitive abilities in order to gain an advantage [20]. The safest options seem to be herbal medicines available in the form of dietary supplements, which contain vitamins, minerals, fatty acids, and other ingredients. According to research, herbs that show procognitive effects include *Ginkgo biloba*, *Siberian ginseng*, *Rhodiola rosea*, Brahmi rasayana, *Mucuna pruriens*, Royal Jelly, caffeine, and curcumin [197].

## 4. Discussion

Esports is a rapidly developing discipline that requires the selection of appropriate nutritional solutions, as well as a holistic approach for the player, to ensure that the player maintains appropriate physical and mental conditioning, both during and after the player’s competitive period. Esports athletes want to improve their skills in the selected games in which they specialize and often spend several hours a day in a forced position, usually in front of computer screens [16]. Therefore, such athletes are exposed to the risk of deteriorating health; vision problems; and pain in the neck, back, or wrists [198]. From a mental perspective, esports athletes are also at high risk of developing depression and burnout symptoms [199]. Regular training can, moreover, lead to addiction, personal hygiene problems, sleep disorders, and social anxiety [198], which can negatively affect the physical, cognitive, and mental health of esports athletes. Thus, appropriate actions should be taken to counteract the negative effects of training.

The training of professional athletes is based on patterns drawn from scientific studies; however, the training involved in esports training is not yet adequately described in the literature. To develop their skills in esports, e-athletes use training in various areas related to overcoming obstacles, teamwork, or individual physical effort, including the cognitive, psychological, physical, technical, and tactical domains [19]. There is a special need for video game players to take care of their cognitive functions due to the competitive practices of video games, which require motor skills, including aiming and manual dexterity [200]; cognitive motor speed, i.e., reaction time and speed of action [200,201]; memory; intelligence [202]; concentration [203]; visual–spatial attention [204]; and many other skills. Understanding the importance of factors influencing brain function and cognitive performance can be critical to athletic performance and the well-being of esports athletes. It seems that a proper diet, energy balance (resulting in an appropriate body structure), an optimal level of physical efficiency and physical activity, and a good state of physical and mental health can help ensure the proper preparation of an esports competitor for competition.

There is a lot of research available on the effects of dietary ingredients on cognition, brain activity, reaction time, and working memory. There have also been many studies focused on the effects of nutrients in preventing or relieving symptoms of neurodegenerative diseases. The mechanisms of the influence of dietary components on cognitive functions are known, but there is a lack of well-developed studies that would assess the relationship between the effect of diet and improvement in esports performance.

### 4.1. The Potential Role of Minerals in Esports

Iodine is a good solution for the proper functioning of the body (i.e., brain function and target hormones) and can improve cognitive functions at every stage of life [43,44,45,46,47,48]. Iodine deficiency may also correlate with a lower IQ index in children and adolescents [49], as well as cognitive declines among school-age children [50,51]. On the other hand, in adults, iodine affects the proper functioning of the thyroid gland, which in turn correlates with an improvement in cognitive functions, such as memory and attention, as well as a proper mood [62,63,64]. Moreover, the development of skills necessary for improving in esports is important during adolescence because, according to current reports, people over twenty years of age have a reduced reaction rate [8]; thus, it is worth considering the appropriate dietary supply of iodine or possible iodine supplementation in the event of a deficiency.

Both an iron deficiency and excess iron may contribute to inadequate functioning of the brain and the formation of cognitive disorders related to difficulties in visual–spatial processing, longer reaction times, and impaired attention and concentration [45,54,58,59,70,71,72,73,74]. All these features are of great importance in esports. Iron supplementation improves general cognitive function, IQ, and measures of attention and concentration in children with anemia [55], and the consumption of iron-fortified foods in adolescents was shown to improve attention, reaction time, and memory [56]. Iron supplementation in middle-aged women was found to improve mood and cognitive functions [57]. Therefore, as with iodine, it is worth ensuring an adequate supply of iron in one’s diet or, in the event of a deficiency, implementing iron supplementation due to the work requirements of e-athletes, which include high concentration, good memory, fast reaction time, and ease of visual–spatial processing. These features are particularly important in multiplayer online battlefield arena games (including LOL, DOTA2) [10,11,14,205,206] and first-person shooters (Overwatch, Counter Strike, Global Offensive CS: GO) [11,14,205,207].

Zinc is a mineral essential for neurogenesis in the brain [76] and is important in the context of learning and memory, as well as emotional and mood control [75]. Zinc supplementation in adults improves spatial working memory [32]. As emphasized by Kowal et al., memory, attention control, and inhibitory skills are particularly important in multiplayer online battlefield arena games and first-person shooters [11]. Hence, it is important to supply zinc within one’s diet in order to cover the daily requirements for this ingredient. In the event of a deficiency, zinc supplementation may be beneficial in improving executive functions and motor development [38], which are also used in multiplayer online battlefield arena games and first-person shooters [11,206].

Esports athletes are a group of players with a high mental load, accompanied by various emotional experiences [208]. The characteristics of esports indicate that the mental condition of players may deteriorate, leading to depression and burnout symptoms, along with deterioration in the psychosocial sphere due to social anxiety, sleep disorders, and problems with personal hygiene [16]. The results of the present research indicate that magnesium deficiency may contribute to weakening of the body, manifested, e.g., by dizziness, fatigue, lethargy, lightheadedness, and loss of appetite [83,84]. Additionally, magnesium deficiency may lead to the formation of oxidative stress, thereby increasing the risk of depression, as well as problems with concentration, working memory, and attention, which are required for gaming [61,82]. Therefore, it is worth ensuring an adequate supply of magnesium in the diets of e-athletes to prevent the negative effects of a deficiency of this element.

### 4.2. The Potential Role of Vitamins in Esports

B vitamins are essential for the proper functioning of the nervous system and the brain, including energy production, DNA and RNA synthesis and repair, genomic and non-genomic methylation, and the synthesis of numerous neurotransmitters and signaling molecules [85]. Nutritional deficiencies in micronutrients such as vitamin B12, folate, and zinc can cause symptoms of depression and dementia, including depressed mood, fatigue, cognitive decline, and irritability [76,107]. The supply of B vitamins within one’s diet, which can cover the demand for these ingredients, has a positive effect on cognitive functions in children and adolescents [95,96]. Moreover, a lower level of vitamin B12 was correlated with deterioration in academic performance and mental and social development, as well as weaker short-term memory and attention in older children or adolescents [97]. Moreover, a properly balanced diet rich in B vitamins may contribute to an improvement in cognitive functions at a later age [94]. The importance of B vitamins in one’s diet should also be emphasized in e-athletes, which represent a group that needs particular support for cognitive functions and proper functioning of the nervous system.

Vitamin E has antioxidant activities [108] that exert a positive influence on cognitive functions [209,210]. Thus, the diets of e-athletes should be rich in vitamin E due to stress, which is a common factor among athletes. Research results indicate that low vitamin E concentrations in the elderly reduce memory performance [109], but another study did not show a significant effect of vitamin E on the cognitive functions of middle-aged and elderly people [99]. There remains a lack of studies that assess the cognitive abilities of healthy young and middle-aged people.

Vitamin D has many important functions in the body, including the modulation of cell growth, neurogenesis, neuroprotection, detoxification, immune function, and reduction in inflammation [110]. Because supplying vitamin D within one’s diet is difficult, and skin synthesis as a result of contact with UV radiation requires special conditions (appropriate clothing enabling skin synthesis, no sunscreen on the skin, and the right time depending on latitude and season) [211], e-athletes are recommended to engage in vitamin D supplementation. Elevated serum vitamin D levels are important for executive functions, such as non-verbal memory [35]. The supplementation and optimal concentration of vitamin D in blood serum may, but does not have to, positively influence cognitive functions, and current scientific reports are contradictory [102,103,104]. However, due to the functions of vitamin D in the body, maintaining an appropriate vitamin D level is necessary for the proper development and functioning of the body. Vitamin D supplementation is especially important for improving mental health. The research results indicate that these components have a beneficial effect on the mental health of healthy people, and meta-analyses confirm the effectiveness of supplementation in reducing depression symptoms [211,212,213,214] and relieving negative emotions [215]. Due to the high mental burden of esports players, vitamin D supplementation seems to be necessary [199].

Vitamin C is an essential component for the functioning of the nervous system due to its modulation of neurotransmitter receptors, brain cell structures, and the synthesis of glial cells and myelin [114,115]. Due to the antioxidant effects of vitamin C and the positive effects of antioxidant components on cognitive functions, the diets of e-athletes should be rich in vitamin C due to stress, which is a factor that often occurs in athletes.

### 4.3. The Potential Role of Dietary Supplements in Esports

Due to the well-documented effect of caffeine in sport, especially in the context of reducing fatigue [120], and improving cognitive functions, such as alertness, concentration, attention, and reaction time [118,120], as well as cognitive performance, motor skills, and physical fitness [117], this supplement is recommended by nutritionists and coaches and used by athletes in esports. To date, two studies on caffeine have been conducted in esports, one involving nine players and the other involving fifteen players [20,152].

Thomas et al. conducted a randomized, double-blind, placebo-controlled, crossover study of nine elite League of Legends players to investigate the cognitive and physical changes associated with consuming an energy drink. The main outcomes included measures of attention, reaction time, and working memory, while the secondary outcomes were based on fatigue. On the basis of the obtained results, it can be concluded that elite esports athletes did not show mental or physical improvement in their results due to the introduced supplementation [20].

Sainz et al. also conducted a study to determine the effects of caffeine on esports performance. A double-blind, cross-randomized, experimental trial was conducted on fifteen professional e-athletes. The aim of this study was to determine the effects of caffeine consumption at a dose of 3 mg/kg b.w. by measuring the simple reaction time during the color test and the hit accuracy and reaction time while playing a first-person shooter. Caffeine consumption at a dose of 3 mg/kg b.w. also improved the accuracy of hitting the target in a first-person shooter among professional e-players [152]. There remains a need for randomized controlled trials with larger study groups. In addition to requiring strong reaction times, esports demands that players be particularly focused, concentrated, and vigilant, necessitating the implementation of all possible measures to improve these skills. Caffeine appears to be a supplement with potentially beneficial effects in esports, and its use may be particularly significant in multiplayer online battlefield arena games and first-person shooters [11,206].

L-Theanine has a positive effect on cognitive functions. When combined with caffeine, L-theanine can improve reaction time, working memory, and the accuracy of task verification [121]; L-theanine also has beneficial effects on concentration, emotional state [122], and attention and can suppress distraction [123]. The combination of a low dose of caffeine with L-theanine and tyrosine may improve athletes’ accuracy of movement and reaction time during strenuous exercise [125]. Due to the aforementioned factors, the use of L-theanine in esports may be a legitimate practice that brings benefits to the player. Training in esports requires attention, memory and focus, accuracy of movement, and quick reaction times from the player. Therefore, it is worth considering L-theanine supplementation, preferably in combination with caffeine, especially for e-athletes who play in multiplayer online battlefield arena games (including LOL, DOTA2) [10,11,14,205,206] and first-person shooters (Overwatch, Counter Strike, Global Offensive CS: GO) [11,14,205,207].

To maintain proper cognitive functions at a high level, one’s diet should be rich in antioxidant ingredients, specifically polyphenols, which have positive effects on cognition and memory and reduce neuronal damage [162,163,164]. The dietary intake of polyphenols can improve visual processing speed, alertness, working memory, and executive functions [128]. Polyphenols should thus be taken into account by e-athletes when planning their diets due to their stressful lifestyles caused by competition. Additionally, polyphenols seem to improve accuracy and speed of attention, which are especially important in both multiplayer online battlefield arena games (including LOL, DOTA2) [10,11,14,205,206] and first-person shooters (Overwatch, Counter Strike, Global Offensive CS: GO) [11,14,205,207].

Cocoa flavanols may also improve cognitive functions, including episodic verbal memory [114]. Due to the positive effect of flavanols on cognitive functions, it is worth introducing products with a high cocoa content into the diets of e-athletes. The addition of flavanols to the diets of e-athletes could be used in multiplayer online battlefield arena games (including LOL, DOTA2) and first-person shooters (Overwatch, Counter Strike, Global Offensive CS: GO) due to the need to develop working memory [11]. Playing for a few hours in front of a computer screen may cause fatigue and sleep disturbances. Screen light can influence the natural circadian rhythm, and this in turn can also influence sleep behavior [216]. The possible impairment of e-athletes’ attention due to the above aspects could potentially be compensated by supplying of cocoa flavanols within athletes’ diets [130].

Beet juice may also have positive effects on cognition and reaction time during prolonged repetitive exercise [132]. The player’s reaction time is particularly important in multiplayer online battlefield arena games (including LOL, DOTA2) [11] and first-person shooters (Overwatch, Counter Strike, Global Offensive CS: GO) [11,14,207]. Therefore, e-athletes should include beetroot juice in their diets to improve cognitive function and reaction time, especially during prolonged training sessions and tournaments.

Creatine supplementation reduces mental fatigue; improves IQ scores; improves working memory performance [135,137] and reaction speed [137]; and affects the state of mood. The use of creatine is recommended in stressful situations, as well as when one’s amount of sleep is limited, due to creatine’s compensatory effects on the cognitive effects of sleep deprivation [138,142,143,144]. The use of creatine supplementation among vegetarians resulted in better memory compared with results in the group consuming meat [140]. The described features are important in esports, especially in multiplayer online battlefield arena games (including LOL, DOTA2) [10,11,14,205,206] and first-person shooters (Overwatch, Counter Strike, Global Offensive CS: GO) [11,14,205,207]. Due to the potential sleep disorders and deficiencies among esports players, in addition to their stressful lifestyles, it is worth considering the use of creatine in esports.

Esports athletes are exposed to mental deterioration, which is associated with the high stress resulting from the competitiveness of esports. Such athletes also face increased social attention and pressure at competitions [217,218]. Thus, the use of probiotic therapy would be beneficial to reduce the negative effects of stress on the structure of the microbiota [149], which, thanks to its immunomodulatory properties, affects the functions of the brain [189]. The use of multi-strain probiotic therapy can modulate functional activity in areas of the brain related to cognitive processes such as problem solving, reasoning, attention, decision-making, learning, and creativity [145]. These features are especially important in multiplayer online battlefield arena games (including LOL, DOTA2) [10,11,14,205,206] and first-person shooters (Overwatch, Counter Strike, Global Offensive CS: GO) [11,14,205,207].

Lutein, due to its anti-inflammatory functions and beneficial effect on eyesight [151], may be a useful ingredient in the diet of e-athletes who spend many hours in front of computer screens [16]. In a study conducted by Ma et al., eyesight improved in healthy people who supplemented with lutein. In particular, the sensitivity to contrast was improved, which may mean that a higher supply of lutein may have a positive effect on visual efficiency [153] (Table 3). Supplements that should be considered in the diet of e-athletes include nootropic substances. Their mechanism contributes to the strengthening of mental functions such as memory, creativity, motivation, attention [196], concentration, and speed [197], which are important from the point of view of esports. The competitive nature of esports causes players to be guided by the choice of various nootropics or drugs that improve cognitive abilities in order to gain an advantage [20]. The safest here seem to be herbal medicines available in the form of dietary supplements, which contain vitamins, minerals, fatty acids, and other ingredients. According to research, herbs that show a procognitive effect are, among others: *Ginkgo biloba*, *Siberian ginseng*, *Rhodiola rosea*, Brahmi rasayana, *Mucuna pruriens*, Royal Jelly, caffeine, and curcumin [197].

In cooperation with e-athletes, it is worth relying on dietary ingredients and supplements that have documented effects in the scientific literature. These supplements include lutein, caffeine, creatine, beetroot juice, and probiotics. In addition to the listed ingredients, there is a need for vitamin D supplementation, especially under limited access to sunlight, as it is impossible to meet the demand for this ingredient within one’s daily diet. Every effort should be made to ensure that the dietary supply of minerals and vitamins does not lead to deficiencies, and the level of ingredients that may be deficient (due to illness, diet, or lifestyle) should be regularly controlled through appropriate tests in order to introduce appropriate supplementation.

### 4.4. Health Behaviors as an Essential Part of Esports Lifestyle

Spending a long period of time sitting position in front of a TV or computer screen may be associated with weight gain and the development of obesity, especially abdominal obesity [219,220,221]. The development of abdominal obesity among those who play video games is particularly worrying due to the associated increased risk of cardiovascular diseases [221,222,223]. The results of studies conducted on the participation of e-athletes indicate that the majority of players either have adequate body weight or are overweight [4,9]. Among the studied esports athletes, there was also a group of people with type 2 and 3 obesity, which suggests that it is necessary to develop appropriate recommendations to facilitate weight reduction, as well as a permanent change in eating habits among this group of players [9]. To counteract the development of overweight and obesity among esports players, nutritional education should be utilized, thereby promoting healthy eating habits and the consumption of balanced meals.

### 4.5. Diet and Nutritional Habits

Proper nutrition is a key factor influencing physical performance [38] and improving mental health. Inadequate diet can lead to an increased risk of obesity [224] and the development of other disorders such as diabetes, cardiovascular diseases, or hypertension [225]. Scientific reports increasingly suggest that health-promoting behaviors such as physical activity, eating a Mediterranean-style diet, and cognitive training protect cognitive functions [226,227,228]. The Mediterranean diet is suggested to have a positive effect on cognitive functions and episodic memory and is associated with a lower risk of cognitive disorders and neurodegenerative diseases [38]. There is, moreover, a relationship between the consumption of vegetables and fruits and an improvement in cognitive functions, especially memory, resulting from the ingredients with antioxidant properties contained in those foods [229]. On the other hand, the influence of the Western diet, which is rich in saturated fatty acids, sugar, and proteins, contributes to the development of diet-dependent diseases such as type 2 diabetes, obesity, dementia, and depression. The Western diet also impairs cognitive abilities. Following the guidelines of the Mediterranean diet thus seems to be an important element of care for the health of e-athletes who seek to improve their cognitive functions. A well-rounded diet includes balanced meals that provide essential nutrients such as proteins, fats, and carbohydrates [38].

An important practice that should be implemented by e-athletes is eating breakfast. Skipping breakfast may lead to a shortened reaction time and short-term memory, in addition to weakening executive functions due to inadequate blood glucose levels [230,231,232]. Every day, e-athletes should also pay attention to adequate hydration, which significantly affects cognitive functions. Dehydration in a situation where water loss exceeds 2% of one’s body weight worsens cognitive performance, attention, executive functions, and coordination [38,233]. Previous studies confirmed that e-athletes eagerly use stimulants such as energy drinks to improve their performance.

The studies conducted so far show that there is no relationship between the consumption of energy drinks and significant improvements in performance [20]. Alcohol is consumed most often by those with a low level of play [234].

### 4.6. Sleep

The quality and duration of sleep also significantly affect the regeneration of athletes. These factors also have an impact on the brain’s performance and ability to cope with immune, emotional, neurological, and psychological factors [38]. Sleep is thus a significant factor that can affect mental health and performance in esports. Scientific research confirms that sleep deprivation negatively affects various cognitive functions and mood [235], thus reducing the chance of victory [236]. E-athletes are particularly vulnerable to lack of sleep due to their many hours of training, high consumption of caffeinated drinks, travel (and the related changes in time zones), and the competitive nature of the discipline [235]. As light-emitting devices are the primary tools in esports, there is an increased risk of sleep disturbances among e-athletes compared with traditional athletes. Existing studies on the general population suggest that the use of electronic devices in the evening hours may cause disturbances in melatonin secretion and thus impair sleep quality and performance during the day [237].

### 4.7. Physical Activity

Physical activity can contribute to meeting the physical requirements of esports. A previous study found that physical exercise can help improve long-term memory, learning, and the ability to acquire motor skills even in the case of extreme effort. Moreover, the effect of exercise is related to an improvement in neural performance in the prefrontal cortex and promotes neurogenesis and plasticity of the brain. The available evidence suggests that physical activity among young people can support cognitive functions and provide such individuals with a competitive advantage, especially in a professional esports career [23]. A study by Trotter et al. indicates that the highest performing e-athletes are also more physically active, with the top 10% of players being significantly more physically active than the bottom 90% of players [9]. Additionally, regular physical activity can be an effective remedy in the treatment of symptoms of depression and stress-induced disorders [238].

### 4.8. Research Limitations

One limitation of the study is that this review is not a systematic review. Considering the currently available literature related to nutrition, there is a need for randomized controlled trials with esports players. Along with the development of scientific research in this area, which may lead to studying the effects of dietary ingredients and supplements on the cognitive abilities of e-athletes, it is worth preparing statistical reviews and meta-analyses.

## 5. Conclusions

Micronutrients are important to optimize cognitive performance and prevent brain disease. A deficit in micronutrients may impair cognitive functions, which are important at every stage of life.Providing vitamin D within one’s diet is problematic, and skin synthesis as a result of UV radiation requires one to meet many restrictive conditions. Thus, supplementation is important to prevent deficiencies and take care of the mental health of e-athletes.Caffeine is a supplement with potentially beneficial effects in esports due to its documented abilities to reduce fatigue and improve cognitive functions such as alertness, concentration, attention, reaction time, cognitive performance, motor skills, and physical fitness.L-Theanine in combination with caffeine can positively impact features such as reaction time, working memory, attention, concentration, and emotional state among e-athletes.The dietary intake of polyphenols can improve alertness, accuracy, speed of visual attention, working memory, and executive functions. Thus, supplementation with polyphenols should be considered by e-athletes due to the stressful and competitive lifestyle of esports.The possible distraction of an e-athlete related to long-term exposure to blue light from the computer screen and the resulting fatigue, disturbances in circadian rhythm, and sleep could potentially be compensated for by supplying cocoa flavanols and creatine within one’s diet. For the sake of players’ eyesight, it would be useful to also use lutein, which, in addition to improving vision and protecting against eye diseases, can also affect cognitive functions.Drinking beetroot juice can potentially improve cognitive performance when performing tasks.Creatine supplementation can affect mood; reduce mental fatigue; and improve intelligence test results, working memory performance, and reaction speed. Thus, creatine could also be potentially used by e-athletes.The use of probiotics seems to support cognitive functions due to the reduction in stress, which negatively affects working memory.There are many supplements (L-theanine, polyphenols, beetroot juice) and plant ingredients (ginseng, mint, *Ginkgo biloba*, *Bacopa monnieri*, lion’s mane, *Rhodiola rosea*, guarana, rosemary, saffron, turmeric, ashwagandha, xanthines) that require further research regarding their effects on cognitive functions.There is a lack of well-designed studies that have investigated the relationship between dietary effects and improved performance in esports. The information presented in this review could, in the future, be used to create specific esports nutritional recommendations that are currently lacking. Therefore, there is a need for randomized controlled trials with esports players.To be successful, esports players need to be in top form, which requires cognitive, physical, and mental support. The most important factor for supporting the abilities of esports players is a rational and proper diet with adequate hydration.

## Figures and Tables

**Table 1 healthcare-10-00186-t001:** Impact of minerals on cognitive function [32,49,50,51,52,53,54,55,56,57,58,59,60,61].

Factor	Subjects and Methods	Results	Conclusion	References
Iodine	Cross-sectional study; 1221 children (6–16 years)	IQ was higher in children with urine iodine >100 μg/L. The intake of non-iodized salt and drinking milk less than once a day was related to the risk of having an IQ below the 25th percentile. The risk of having an IQ below 70 was greater in children with urinary iodine levels <100 μg/L.	Intake of iodine in children in the developed world may affect IQ. The IQ of children with mild iodine deficiency can be improved by adequate iodine intake through the diet until the urinary iodine concentration is above 100 μg/L.	[49]
	Randomized, placebo-controlled, double-blind trial; 184 children (10–13 years)	Iodine supplementation improved scores for 2 of the 4 cognitive subtests (picture concepts and matrix reasoning but not for letter-number sequencing or symbol search).	Mild iodine deficiency could prevent children from attaining their full intellectual potential. Iodine supplementation in children with deficiency improves perceptual reasoning.	[51]
	Randomized, placebo-controlled, double-blind intervention trial; 310 children (10–12 years)	Iodine treatment improved performance on 4 of 7 tests: rapid target marking, symbol search, rapid object naming, and Raven’s Coloured Progressive Matrices compared with placebo.	Iodine supplementation in schoolchildren with moderate deficiency improves information processing, motor skills, and visual problem solving.	[50]
Iron	Review	Children and young adults who had iron deficiency anemia in infancy showed poorer inhibitory control and executive functioning as assessed by neurocognitive tasks.	There is a need to prevent iron deficiency in infancy due to persistent poorer cognitive, motor, affective, and sensory system functioning.	[52]
	Systematic review and meta-analysis; 32 studies including 7089 children (5–12 years)	Iron supplementation improved global cognitive scores, intelligence quotient among anemic children and measures of attention and concentration.	In young players with anemia, it is worth using iron supplementation to improve cognitive functions.	[55]
	Systematic review; 10 randomized controlled trials and 1 non-randomized controlled trial; women (12–55 years)	Improvement in aspects of mood and cognition after iron supplementation (in 7 studies). Iron supplementation appeared to improve memory and intellectual ability in participants, regardless of whether the participant was initially iron insufficient or iron-deficient with anemia.	Adequate iron levels are a key factor in intellectual performance.	[57]
	Review; 89 studies	Iron deficiency had a negative impact on cognition, behavior, and motor skills.	There is some evidence that iron supplementation improves cognition.	[54]
	Observational study, 39 children (7–11 years)	Positive relationship between spatial IQ and mean iron content in the basal ganglia and in the caudate specifically.	Iron content in specific regions of the iron-rich deep nuclei of the basal ganglia influences spatial intelligence.	[53]
	Double-blind, randomized, intervention study; 140 Indian boys (12–16 years)	Daily iron intake from pearl millet was higher in those consuming biofortified (19.6 mg/d) compared with conventional pearl millet (4.8 mg/d). Compared with conventional pearl millet, the consumption of biofortified pearl millet resulted in greater improvement in attention and memory.	Consuming iron-biofortified pearl millet improves iron status and some measures of cognitive performance in Indian adolescents.	[56]
	Cohort trial; 428 adolescents (12.0 ± 0.4 years) from China	Iron deficiency was associated with slower performance of tasks that measured abstraction, mental flexibility, and spatial processing capacity. High serum iron levels were associated with lower accuracy in the spatial processing ability task and longer reaction time in the abstraction and mental flexibility task compared with normal levels.	Both iron deficiency and high iron levels lead to decreased neurocognitive performance in a domain-specific manner in early teens.	[58]
	Review; 41 human studies	Increased iron levels in caudate nuclei correlated with worse memory and general cognitive performance in adulthood. The increase in iron levels in the hippocampus and thalamus was associated with poorer memory performance. The increased iron level in the putamen and globus pallidus affects general cognition.	Brain iron is detrimental to cognitive health.	[59]
Zinc	Randomized double-blind placebo-controlled trial; 387 healthy adults aged 55–87 years	3-month zinc supplementation at a dose of 15 or 30 mg/day had a positive effect on spatial working memory, while a dose of 15 mg/day had a negative effect on the measurement of attention. Younger adults (<70 years old) performed significantly better on all tests than older adults (>70 years old).	Zinc supplementation can have a positive effect on spatial working memory.	[32]
	Systematic review and meta-analysis; 18 studies; 12 randomized controlled trials (11 in children and 1 in adults) and 6 observational studies (2 in children and 4 in adults)	A correlation between zinc intake or status with one or more measure of cognitive function was shown (9 of analyzed 18 studies). Meta-analysis of data from the adult studies was not possible because of the limited number of studies.	There is no significant effect of zinc supplementation on cognition in children. Supplementation may improve executive functions and motor development.	[60]
Magnesium	Systematic review; 32 articles	Magnesium supplementation showed positive results in depressive symptoms (12 studies). The decreased level of magnesium in the plasma correlated with the occurrence of depression, assessed using psychometric scales (7 studies). Consumption of magnesium improved the symptoms of depression (2 studies). The consumption of magnesium in combination with antidepressants improved the symptoms of depression (2 studies). Concentration of magnesium in the raw material had no significant influence on the occurrence of panic or anxiety disorders (2 studies). Higher serum magnesium levels in depressed and stressed patients correlated with a lower Hamilton anxiety score (2 studies). Low levels of magnesium correlated with attention deficit hyperactivity disorder (ADHD) (2 studies). Lower levels of magnesium were observed in autism spectrum disorder (1 out of 3 studies). Eating disorders and schizophrenia were associated with differences in magnesium levels in some aspects of the disease.	Supplementation with magnesium could be beneficial in mental disorders.	[61]

**Table 2 healthcare-10-00186-t002:** Impact of vitamins on cognitive function [35,92,93,94,95,96,97,98,99,100,101,102,103,104,105].

Factor	Subjects and Methods	Results	Conclusion	References
B vitamins	Systematic review and meta-analysis of 31 English-language, randomized placebo-controlled trials	B-vitamin supplementation did not show an improvement in Mini-Mental State Examination scores for individuals with and without cognitive impairment compared with placebo.	Elevated plasma homocysteine levels increase the risk of cognitive impairment and dementia. There is no clear evidence confirming the beneficial effect on cognitive functions resulting from lowering the level of homocysteine with B vitamins (heterogeneity of studies).	[93]
	Multicenter study; 1398 blacks and 1738 whites (25.1 ± 3.6 years)	Higher consumption of vitamin B6 and vitamin B12 correlated with better psychomotor speed assessed by Digit Symbol Substitution Test (DSST) scores.	Higher intake of B vitamins in early adult years was associated with improved cognitive function in later adult life.	[94]
	Study; 317 children (11.8 ± 3.3 years)	Vitamin B1 consumption had a positive effect on the results of the numerical tasks and modalities of symbolic digits (neurocognitive tests). The consumption of vitamin B6 showed a positive correlation with the results of the digit range tasks. The consumption of vitamins B1, B2, B6, and niacin was negatively correlated with omission errors, which indicate inattention.	A diet high in B vitamins correlates with better neurocognitive test results. Diet is closely related to the cognitive functions of healthy children and adolescents.	[95]
	Study; 72 adolescents (10–16 years)	Controls who followed the standard diet from birth without eliminating meat products scored better on most psychological tests compared with those following the macrobiotic diet with low or normal cobalamin status.	Cobalamin deficiency without hematologic signs can possibly cause impaired cognitive performance in adolescents.	[96]
	Review; 17 studies: 3 cross-sectional, 1 case–control, and 12 cohort studies and 1 randomized trial.	Observational studies to date have demonstrated associations between vitamin B-12 status or dietary intake and cognitive outcomes in children.	A diet rich in vitamin B12 is recommended in adolescence due to its positive influence on cognitive functions.	[97]
	Randomized, double blind, placebo controlled study; 818 adults (50–70 years)	3 years of folic acid supplementation improved memory, information processing speed, and sensorimotor speed compared with the placebo group.	3 years of folic acid supplemented intake significantly improved cognitive functions that most often decline with age.	[92]
	Systematic Review, Meta-Analysis, and Meta-Regression; 53 randomized trials	No evidence for an effect of B12 alone or B complex supplementation on any subdomain of cognitive function outcomes.	Vitamin B12 supplementation possibly does not improve cognitive function or depressive symptoms in patients without advanced neurological disorders.	[98]
Vitamin E	Systematic Review and Meta-Analysis; 24 trials (6 omega-3 fatty acids, 7 B vitamins, 3 vitamin E, 8 other interventions)	The vitamin E studies did not show any meaningful impact on cognitive outcomes.	Vitamin E supplementation had no effect on cognition in mentally healthy middle-aged and older adults.	[99]
	Meta-analysis; 5 trials (vitamin B); 1 trial vitamin E; 1 trial (vitamin C and E)	No significant effect of three years of supplementation with vitamin E (1000 IU of alpha-tocopherol twice daily) on overall cognitive function, episodic memory, speed of processing, clinical global impression, functional performance, adverse events, or mortality.	Three years of major vitamin E supplementing suggested scarce possibilities of reduced risk of progression to dementia.	[100]
Vitamin D	Mendelian randomization study; 17 cohorts: 172349 participants	Associations of serum [25(OH)D] with global and memory-related cognitive function were non-linear (lower cognitive scores for both low and high [25(OH)D].	No evidence for serum [25(OH)D] concentration as a causal factor for cognitive performance in mid- to later life.	[101]
	Systematic review and meta-analysis; 17 articles	Vitamin D deficiency (<25 nmol /L or 7–28 nmol/L) was associated with an increased risk of dementia compared with people with sufficient vitamin D intake (≥50 nmol/L or 54–159 nmol/L) (meta-analysis of 5 studies).	Low vitamin D levels might contribute to the development of dementia.	[102]
	Systematic review and random effect meta-analysis; 26 observational and 3 intervention studies	Low vitamin D was associated with deteriorated cognitive performance and cognitive decline; with cross-sectional studies yielding a stronger effect compared with longitudinal studies. Vitamin D supplementation showed no significant benefit on cognition compared with control.	Low vitamin D levels deteriorate cognitive functions. There is no clear benefit from vitamin D supplementation.	[103]
	Randomized placebo controlled trial; 128 participants (at least 18 years of age)	Vitamin D supplementation (5000 IU of cholecalciferol) did not cause changes in working memory, inhibition of reaction, cognitive flexibility, or secondary symptoms (predisposition to hallucinations, psychotic experiences, and the assessment of depression, anxiety, and anger) despite a significant increase in serum vitamin D levels.	Vitamin D supplementation has no impact on cognitive or emotional functioning in healthy young adults.	[104]
	Randomized trial; 82 healthy adults	Supplementation with a high dose of vitamin D (4000 IU/d) improved the efficiency of non-verbal (visual–spatial) memory, and in the group with lower baseline [25(OH)D] levels (<75 nmol/L), it improved significantly.	A higher level of [25(OH)D] is especially important for a higher level of cognitive functioning, especially non-verbal (visual) memory, which also uses executive processes.	[35]
Vitamin C	Systematic Review; 50 studies, with randomized controlled trials (RCTs, *n* = 5), prospective (*n* = 24), cross-sectional (*n* = 17), and case–control (*n* = 4) studies	Studies demonstrated higher mean vitamin C concentrations in the cognitively intact groups of participants compared with cognitively impaired groups.	No correlation between vitamin C concentrations and cognitive function was apparent in the cognitively impaired individuals.	[105]

**Table 3 healthcare-10-00186-t003:** Impact of supplements on cognitive function [20,117,118,119,120,121,122,123,124,125,126,127,128,129,130,131,132,133,134,135,136,137,138,139,140,141,142,143,144,145,146,147,148,149,150,151,152,153].

Factor	Subjects and Methods	Results	Conclusion	References
Caffeine	Review	Caffeine doses of around 0.5–4.0 mg/kg b.w. (~40–300 mg) improved cognitive function in well-rested individuals, while doses of 3–7 mg/kg b.w. (~200–500 mg) taken approximately 1 h before exercise improved physical performance. The dose–response exhibited high interindividual variability.	Doses of 1–4 mg/kg b.w. improve alertness, concentration, and reaction time, but there is less consistent impact on memory and executive functions, such as assessing the situation and making decisions.	[118]
	International Olympic Committee consensus statement	Doses of 3–6 mg/kg b.w. caffeine improved cognitive, motor skills, and exercise performance in many types of sports.	Caffeine supplementation may improve cognitive, motor skills, and exercise performance.	[117]
	Randomized, double-blind, placebo-controlled crossover trial; 9 elite League of Legends (LoL) esports players	The administration of a supplement (AI Reload (118 mL) that included 150 mg caffeine (1.9 ± 0.3 mg/kg b.w.) designed to improve performance demonstrated no ergogenic effects relative to the indices examined in this study (measures of attention, reaction time, working memory, and fatigue).	Three LoL games do not accumulate mental fatigue in elite LoL players. The administration of a supplement containing caffeine does not have a positive effect on the cognitive functions of LoL players.	[20]
	Crossover, double-blind study; 10 male subjects	Intake of 3 mg/kg of caffeine boosted performance on the Stroop task under both incongruent and congruent conditions and increased mean oxygenated hemoglobin under the congruent condition. Ingestion of 6 mg/kg of caffeine increased efficiency on the Stroop task under the incongruent condition.	Low-dose caffeine intake proved to have greater effects on cognition and brain activation compared with midrange and high caffeine doses, implying that small doses of caffeine may be the supplement of choice in enhancing executive function and prefrontal activities.	[119]
	Double-blind, crossover, randomized experimental trial; 15 professional e-gamers (age = 22 ± 3 years).	The acute ingestion of 3 mg/kg of caffeine improved both reaction time and accuracy in hitting targets.	Caffeine consumption (3 mg/kg b.w.) can be considered an ergogenic aid for esports players (in first person shooters) due to its effect on increasing accuracy and hit time.	[152]
	Systematic Review and Meta-Analysis; 13 studies, of which 5 studies were included in the meta-analysis	Caffeine showed beneficial effects in tasks requiring attention, accuracy, and speed (meta-analysis). Supplementation with a low/moderate dose of caffeine before and/or during exercise helped raise mood, energy, and attention, but also enhanced simple reaction and response time, as well as augmenting memory and easing fatigue, though this may have depended on test protocols (13 studies).	Caffeine supplementation in sports requiring attention and focus may be considered, but more research is needed.	[120]
Combination of caffeine and theanine	Randomized, placebo-controlled, double-blind, balanced crossover study; 24 participants (9 male and 15 female, mean age 21.3 years)	A combination of 250 mg of L-theanine and 150 mg of caffeine improved reaction time, working memory, and accuracy of task verification.	The combination of caffeine and L-theanine seems to be justified in sports that require quick response, memory, and accuracy.	[121]
	Systematic review and meta-analysis (11 randomized placebo-controlled human studies)	The combination of L-theanine with caffeine increased alertness, increased the accuracy of switching attention, slightly increased the accuracy of non-sensory visual attention, and slightly increased the accuracy of non-sensory auditory attention.	Caffeine in combination with L-theanine had a beneficial effect on the cognitive function and mood.	[122]
	Systematic review (49 studies)	Caffeine in a dose of 40 mg improved performance in demanding long-term cognitive tasks as well as self-reported alertness, agitation, and vigor. L-Theanine alone improved relaxation, tension and calmness reported by patients themselves, starting with 200 mg. The combination of L-theanine and caffeine improved performance in attention-switching and alertness tasks, but to a lesser extent than caffeine itself.	L-Theanine and caffeine both have apparent advantageous effects on continued alertness, memory, and improved attention. L-Theanine helped to relax by curbing arousal stimulated by caffeine.	[123]
	Systematic review of controlled trials, crossover studies cross-sectional studies, and cohort study (21 studies)	Green tea influenced psychopathological symptoms (e.g., reduction in anxiety), cognition (e.g., benefits in memory and attention), and brain function (e.g., activation of working memory seen in functional MRI).	Caffeine in combination with L-theanine has a beneficial effect on cognition, while the separate use of both substances has a smaller effect.	[124]
Combination of caffeine, theanine, and tyrosine	Randomized, double-blind, placebo-controlled crossover trial; 20 current or former male collegiate athletes (age: 20.5 ± 1.4 years)	The combination of a low dose of caffeine with theanine and tyrosine improved the accuracy of athletes’ movements and reaction time during a series of grueling exercises.	Supplementation recommended in exercises demanding accurate movements and quick reaction time.	[125]
Polyphenols	Systematic review and meta-analysis; 18 studies	The acute consumption of polyphenols enhanced processing of fast-paced visual stimuli in young participants.	Acute polyphenol consumption might improve speed in rapid visual information processing task, a higher order task with elements of vigilance, working memory, and executive function, in young participants	[128]
Cocoa	Randomized, double-blind, crossover study; 12 healthy men	At rest, cocoa flavanol intake increased cerebral oxygenation, but not brain-derived neurotrophic factor concentrations, and no impact on executive function was detected.	Positive influence of cocoa flavanol on brain oxygenation during rest was revoked by a strong increase in perfusion and brain oxygenation caused by exertion.	[126]
	98 healthy young adults (*n* = 57 females) aged 18–24 years	Dark chocolate consumption (70% cocoa) was associated with better verbal memory performance for several outcome measures of the Rey Auditory Verbal Learning Test relative to the white chocolate consumption; however, there were no effects on mood.	70% cocoa dark chocolate consumption can benefit verbal episodic memory two hours post consumption in healthy young adults relative to a white chocolate control. A daily serving (35 g) of dark chocolate can benefit the brain of healthy consumers.	[127]
	Review; 11 intervention studies that involved a total of 366 participants	After acute consumption, these beneficial effects seemed to be accompanied by an increase in cerebral blood flow or cerebral blood oxygenation. After chronic intake of cocoa flavanols in young adults, a better cognitive performance was found, together with increased levels of neurotrophins.	The beneficial effect of cocoa flavanols on cognitive function and neuroplasticity was supported and indicates that such benefits are possible in early adulthood.	[129]
	Review, 15 human studies	Regular consumption of flavanols had a neuroprotective effect and also improved cerebrovascular and metabolic functions.	Consuming cocoa flavanols may have beneficial effects in maintaining cognitive performance by improving indicators of general cognition, attention, processing speed, and memory.	[130]
	Systematic Review; 18 studies	Consuming chocolate or cocoa products improved lipid (triglyceride) profiles. The effect of chocolate on all other outcome parameters did not differ significantly (including cognitive functions).	There is no evidence that cocoa and cocoa-containing products may be beneficial for cognition.	[131]
Beetroot juice (nitrate)	Randomized, double-blind, crossover study, 16 male team-sport players	The total work done during the sprints was greater in the nitrate-rich beetroot juice group compared with the placebo group. The response time to cognitive tasks in the second half of the sprint improved in the nitrate-containing beet juice group compared with the placebo group.	Dietary NO_3_^−^ enhances repeated sprint performance and may attenuate the decline in cognitive function (and specifically reaction time) that may occur during prolonged intermittent exercise.	[132]
	Randomized, double-blind, placebo-controlled trial, 40 healthy adults (18–27 years)	Dietary nitrate modulated the hemodynamic response to task performance, with an initial increase in prefrontal cortex cerebral blood flow at the start of the task period, followed by consistent reductions during the least demanding of the three tasks utilized. Cognitive performance was improved on the serial 3 s subtraction task.	Single doses of dietary nitrates may modulate the cerebral blood flow response to task performance and potentially improve cognitive performance.	[133]
	Double-blind, placebo controlled, crossover trial; thirteen younger (18–30 years) and 11 older (50–70 years)	Response time improved in the Stroop test after beetroot juice supplementation for both groups. Acute BR supplementation increased plasma nitrite levels and to a greater extent reduced diastolic BP in the elderly; while systolic BP was lowered in both older and younger subjects.	Acute supplementation with beetroot juice can reduce blood pressure and improve aspects of cognitive performance; thus, having potential health benefits for both younger and older adults.	[134]
Creatine	Double-blind placebo-controlled trial; 24 healthy volunteers (19 men and five women, 24.3 ± 9.1 years old)	After taking the creatine supplement, task-evoked increase in cerebral oxygenated hemoglobin in the brains of subjects measured by near infrared spectroscopy was significantly reduced, which is compatible with increased oxygen utilization in the brain.	Dietary supplement of creatine (8 g/day for 5 days) reduces mental fatigue when subjects repeatedly perform a simple mathematical calculation.	[136]
	Double-blind, placebo-controlled, crossover trial; 25 vegan or vegetarian subjects (12 males (median age of 27.5, range of 19–37 years), 33 females (median age of 24.9, range of 18–40 years)	Oral creatine supplementation (5 g/day for six weeks) improved IQ scores and working memory performance in 45 young adult vegetarians. Creatine supplementation had a positive effect on both working memory (digit spread) and intelligence (Raven’s Advanced Progressive Matrices), both tasks requiring processing speed.	These findings underline a dynamic and significant role of brain energy capacity in influencing brain performance.	[137]
	Clinical trial; volunteer male (*n* = 17) and female (*n* = 3) sports science and adventure education majors (mean age: 21.11 years)	At 24 h, the creatine group (20 g/day for 7 days) demonstrated significantly less change in performance in random movement generation (RMG), choice reaction time, balance, and mood state.	Following 24 h sleep deprivation, creatine supplementation had a positive effect on mood state and tasks that place a heavy stress on the prefrontal cortex.	[138]
	Double-blind, placebo-controlled study; 34 participants (including 12 females); mean age of 21 years (SD: 1.38; range: 18–24).	Creatine ethyl ester supplementation (5 g/day for 15 days) improved cognition on some tasks. Creatine dosing led to an improvement over the placebo condition on several measures.	Although creatine seems to facilitate cognition on some tasks, these results require replication using objective measures of compliance.	[139]
	Randomized, double-blind, placebo-controlled trial; Female undergraduates (*n* = 121), mean age 20·3 (SE 2·1) years, meat-eaters (*n* = 51) and vegan or vegetarian (*n* = 70).	Creatine supplementation (20 g of creatine supplement for 5 d) did not affect the indicators of verbal fluency and alertness. However, in vegetarians, supplementation with creatine resulted in better memory compared with the group consuming meat.	For vegetarians, it is worth using creatine supplementation to improve memory.	[140]
	Systematic review of randomized controlled trials; 6 studies (281 individuals)	Short-term memory and intelligence/reasoning can be improved by administering creatine. Effect of creatine on other cognitive domains, i.e., long-term memory, spatial memory, memory scanning, attention, executive functions, reaction inhibition, word fluency, reaction time, and mental fatigue was not clear cut. Vegetarians responded better than meat eaters to memory tasks, but no differences were observed for other cognitive domains.	Oral creatine administration may improve short-term memory and intelligence/reasoning of healthy individuals, but its effect on other cognitive domains remains unclear. Findings suggest potential benefit for aging and stressed individuals.	[135]
	Pilot trial in healthy men (*n* = 5)	Guanidinoacetic acid (GAA, 3 g·day^−^), a naturally occurring creatine precursor, was reported to have a superior influence on brain creatine content when compared with an equimolar dose of creatine.	GAA as a preferred alternative to creatine for improved bioenergetics in energy-demanding tissues.	[141]
	Blinded, placebo-controlled crossover design; 10 rugby backs (mean ± SD, age; 20 ± 0.5 years)	No fall in skill performance was seen with caffeine doses of 1 or 5 mg/kg, and the two doses were not significantly different in effect. Similarly, no deficit was seen with creatine administration at 50 or 100 mg/kg, and the performance effects were not significantly different.	Creatine can be used in stressful situations in which there may be a temporary decrease in creatine levels and may offset the negative cognitive effects of sleep deprivation.	[142]
	Review	It appears that creatine was most likely to exert an influence in situations whereby cognitive processes were stressed, e.g., during sleep deprivation, experimental hypoxia, or during the performance of more complex, and thus more cognitively demanding tasks.	In situations of sleep deprivation and performing more complex tasks, creatine supplementation may turn out to be beneficial.	[143]
	Review	There was a potential for creatine supplementation to improve cognitive processing, especially under conditions characterized by brain creatine deficits, which could be induced by acute stressors (e.g., exercise, sleep deprivation) or chronic, pathologic conditions (e.g., creatine synthesis enzyme deficiencies, mild traumatic brain injury, aging, Alzheimer’s disease, depression).	The optimal creatine protocol able to increase brain creatine levels is still to be determined.	[144]
Prebiotics, probiotics and fermented food	A meta-analysis of randomized controlled trials; 22 studies (*n* = 1551)	Despite several individual studies (14 of 22) reporting significant improvements in specific cognitive domains, results of the pooled meta-analysis found no significant effect for any intervention for global cognition.	These results do not support the use of probiotic, prebiotic, and fermented food interventions for cognitive outcomes.	[147]
Prebiotics	Narrative review; 14 studies, (5 randomized, crossover trials, 3 double-blind, and 2 nonblinded studies)	Chronic prebiotic interventions (>28 d) improved affect and verbal episodic memory compared with a placebo. Acute prebiotic interventions (<24 h) were more efficient in improving cognitive variables (e.g., verbal episodic memory).	Acute prebiotic interventions (<24 h) can be used to improve cognitive variables. However, more research is needed.	[146]
	Randomized, double-blind, crossover, controlled trial; 18 healthy female participants	Polydextrose improved cognitive flexibility (reducing the number of errors made in the Intra-Extra Dimensional Set Shift task). Better performance in terms of retention of attention was observed due to the greater number of correct answers and rejections in the quickly processing visual information task. Although there was no change in microbial diversity, *Ruminiclostridium 5* abundance increased significantly after polydextrose supplementation compared with placebo.	Supplementation with the polydextrose resulted in a modest improvement in cognitive performance. The results indicate that polydextrose could benefit gut-to-brain communication and modulate behavioral responses.	[148]
Probiotics	Randomized controlled trial; healthy female subjects, (aged 18–40 years)	Subjects with a higher increase in Ruminococcaceae_UCG-003 abundance after probiotics were also more protected from negative effects of stress on working memory after probiotic supplementation.	Gut microbial alterations, modulated through probiotics use, are related to improved cognitive performance in acute stress circumstances.	[149]
	Randomized, double-blind, placebo-controlled trial, 45 right-handed healthy participants (aged 20–40 years)	Functional connectivity changes were observed in the default mode network, salience network, and middle and superior frontal gyrus network in the probiotic group as compared with the placebo and control groups.	The results demonstrated that there is a close relationship between the effects of probiotic intervention on behavioral and neuroimaging readouts.	[145]
Lutein	Review	Dietary sources of lutein (avocado, spinach) had an influence of serum level of lutein and could lead to the accumulation of lutein in retinal neural tissue and may maintain eye and brain health.	Dietary intake and supplementation of lutein can improve cognitive and vision abilities.	[150]
	Randomized controlled trial; 20 healthy subjects	In a study, the elderly (mean age 63) consumed avocados (high source of lutein), which increased MPOD (macular pigment optical density) and was associated with the improvement in cognitive functions.	Dietary intake of lutein has significant impact on the improvement in cognitive functions.	[151]
	Randomized double blind, placebo-controlled trial; 37 healthy subjects (aged 22–30 years)	Lutein supplementation for 12 weeks increased levels of serum lutein and improved contrast sensitivity in both groups.	Higher intake of lutein may be beneficial for visual performance.	[153]

## Data Availability

Not applicable.

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
