# Peer review of "Can Nutrients and Dietary Supplements Potentially Improve Cognitive Performance Also in Esports?"

_healthcare, 2022, doi:10.3390/healthcare10020186_

Round 1

Reviewer 1 Report

Thank you for sending me this paper to review. Although this paper might have some interesting comments about nutrition (though it really reads like a literature review), this paper, honestly, is not really an esports paper in the slightest. This is a nutrition paper with a light veneer of esports gently placed on top – a single paragraph (!!) at the start of the paper, a handful of words in an astonishingly short discussion and conclusion – and that’s it. If this paper wants to do what it claims it wants to do, to assess nutritional needs and potential supplements and so forth for esports players, it needs to actually engage with esports literature and tell us up-front why these things might be relevant to esports; it needs to properly and thoroughly discuss the cognitive and emotional and physical demands of gaming and esports throughout; and it needs to hugely expand its discussion and conclusion.

So, like I say, the main issue is that there’s just nothing about esports here. The entire esports “literature review” is one paragraph (!) containing 5 references, and of those, only #2 and #4 are actually about esports! So we have a literature review of... 2. (Although I also note ref 87 is about esports, so I will say there is a total of 3 esports references). In turn, this is not a perfect metric of course, but if you search for “esports” and “e-sports” (the paper needs to just choose one spelling and stick with it) the words combined only show up 29 times. Almost half of these are on the very first page in the extremely short and sadly totally insufficient esports literature review, and then a handful in the discussion saying “Maybe nutrition is a good idea for esports”, and a handful in the references.

So essentially, *the main body of the paper has nothing to do with esports*. It’s just a list of nutritional supplements, as far as I can tell, and once or twice a few hypotheses about how or why these might have esports value, but even this is not always the case. Just to choose one example, "3.2.3. Polyphenols" is just a section talking about polyphenols. It is mentioned once or twice in these sorts of sections that X or Y might improve reflex speed or visual acuity or whatever... but that doesn’t then tell us anything about the relationship between those abilities and gaming / esports.

Also, “e-sportsmen” should be “esports players” or something similar, both because that’s the used term and to avoid assuming gender.

So, what this paper needs to do is three things.

Firstly, we need some actual engagement with esports literature about the topics the paper is talking about, i.e. the demands of esports in cognitive, emotional, stress, physical, etc, terms. There are lots of these to be found on Google Scholar or wherever, but at the very minimum, this paper should engage with these articles...

Hilvoorde, I. V., & Pot, N. (2016). Embodiment and fundamental motor skills in eSports. Sport, Ethics and Philosophy10(1), 14-27.

Johnson, M. R., & Woodcock, J. (2021). Work, play, and precariousness: An overview of the labour ecosystem of esports. Media, Culture & Society, first published online.

Ferrari S (2013) eSport and the human body: Foundation for a popular aesthetics. In: Proceedings of DiGRA 2013: DeFragging Game Studies. Available at: http://www.digra.org/wp-content/ uploads/digital-library/paper_387.pdf (accessed 19 May 2015)

Brock, T., & Fraser, E. (2018). Is computer gaming a craft? Prehension, practice, and puzzle-solving in gaming labour. Information, Communication & Society, 21(9), 1219-1233.

Witkowski E (2012) On the digital playing field and how we 'do sport' with networked computer games. Games and Culture 7(5): 349-374

...which cover in detail questions of motor function, play skill, labour, stress, mental effort, and so forth, required in esports. These should feature in a significant literature review at the top of the paper, and should (see below) be used throughout, and these are the sorts of articles which this paper should be in conversation with during the discussion and conclusion. I would hope to see at least a full page of esports literature review at the start of this paper, as a minimum.

Secondly, I suggest shortening and compressing this gigantic list of supplements and nutritional choices and so forth quite substantially. In particular I’m not sure what the huge tables really contribute to this paper. With the words freed up, the authors should in each section make it clear HOW and WHY each supplement might be useful to gaming. Some esports games require rapid decisions, some require sitting comfortably for long periods of time, some require complex strategic choices, some require excellent eyesight, some require other things, and so forth. What sorts of esports might each supplement help? Why? Why not others? What might these mean for esports?

Thirdly and lastly, the discussion and conclusion are astoundingly short, and tell us almost nothing. These should be rewritten, working in esports literature, to offer a proper discussion of the potential place of nutrition, supplements, etc, in esports competition. Again, this will require proper engagement with esports literature and the physical, cognitive, stress, labour, emotional, etc, demands of esports.

Overall, therefore, this paper needs quite a bit of work to meet the high ambitions of its (currently misleading) title, “The effect of dietary nutrients on cognitive performance in e-Sports”. The paper doesn’t actually talk about the effect of dietary nutrients on cognitive performance in esports; it instead mentions esports is a thing that exists, and nutrients exist with various benefits. That’s the paper, there’s just no connection whatsoever between the two sections. However, with a thorough engagement with relevant esports literature; a rework of the main sections to explicitly and consistently connect the nutrients being mentioned to esports / gaming; and a rework of the discussion / conclusion sections; this paper might have something interesting. I recommend Major Revisions.

Author Response

Thank you very much for your comments and suggestions in this paper. 

  • In response to the first point of the review we developed the information about esports in the whole article and included actual and recommended literature. We considered all features are demanded of video gaming and potentially in esports too .  At the start of the review we have brought more information in this discipline. 
  • It’s appropriate to highlight the one of our main assumptions was to be the nutrition review. Due to the lack of scientific evidence it might be the establishment to future research of esports players. 
  • Referring to the third point of the review, we broadened the discussion to esports literature. We've also added information on the potential effects of nutrients and supplements in esports and their use in different types of games, depending on the characteristics desired for the game.

  • The main body of this paper focus on nutritional supplements, because each of them might have an influence on the optimization of health condition and through this improve cognitive functions. We can’t draw any conclusions "how and why" each supplement might be useful to esports due to shortage of studies. Therefore, we have made an attempt to discuss micronutrients and dietary supplements and their potential impact on cognitive functions in esports, referring to the characteristics of players.

  • The main aim of presented table was to summarize all the results of researches in one field. 

  • We changed the title of article because of we recognized that asking a question will be more favourable for considerations.

  • Significant changes in our papers: due to the fact that we included more esports work, we had to broaden the search period. We also changed the title, we broadened the discussion and we clarified the purpose of the work.

Reviewer 2 Report

Very interesting paper with proper compilation and structure.

Small suggestions for improvement:

  • line 13 - the abstract is supposed to have references?
  • - line 79 presentation of references
  • -lines 245, 259, 272, ... table 2 is bold? 

Author Response

Thank you very much for your comments and suggestions in this paper. 

  • As suggested, we have removed the link in the abstract (line 13) of the article. It was placed in connection with a reference to our previous work on a similar subject (KarpÄ™cka, E .; FrÄ…czek, B. Macronutrients and water - do they matter in the context of cognitive performance in athletes? Baltic Journal of Health and Physical Activity 2020, 12 , 114-124. DOI: 10.29359 / BJHPA.12.3.11). The article currently being prepared is to be a continuation of the first work.

  • We added a reference to the sentence in the iodine section of the article (line 79).

  • We checked the lines 245, 259, 272 and corrected the references to the tables.We don’t know why the references in the text to the table has changed because we used the same formatting (“Table 1, 2 oraz 3 whithout the name of the table”) in each line. We checked it and corrected.

  • Significant changes in our paper: due to the fact that we included more esports work, we had to broaden the search period. We also changed the title, we broadened the discussion and we clarified the purpose of the work.

Reviewer 3 Report

While there is a clear need to understand the relationship between nutrition and cognitive function for the athletes of e-sports, there is not enough specific literature for a review.  General reviews on nutrients and cognitive health already exist.  To improve future versions, please consider the following:

  • Focus on a single cognitive outcome that is critical for e-sports (or at least define specifically what you are looking for), and perform meta-analyses when possible
  • When listing the search terms, give the exact search criteria: ((nutrition) OR (nutrients) OR (micronutrients) OR (minerals)) AND ((e-sports) OR (e sports) OR (electronic sports)) 
  • Each section should focus on summarizing or synthesizing the papers instead of focusing on the details of specific papers
  • There was a sense that much of the verbiage was written without needing to consider the specific results from the literature search. 

All this to say, there was a lot of effort put into this review - with editing/reframing, this review can still be publised.  

Author Response

Thank you very much for your comments and suggestions in this paper. 

  • As suggested in the discussion, we tried to use the suggested search method, but it did not provide us with relevant results that could be used in the review. Multiple search engine searches have been undertaken to find links between dietary components and cognitive function, as there is currently a lack of papers examining the effects of nutrients on cognitive performance in esports (apart from two papers on caffeine).

  • In response to the review, we broadened the discussion in which we tried to refer to the effects of micronutrients and dietary supplements on cognitive functions, described previously and supported by the literature, to esports, and more specifically to the important features of players in e-sport, in order to propose the direction of future scientific research.

  • We described a potential feature required in esports. There is a lack of evidence in esports field (more specifically, no studies on the effects of diet and dietary components on cognitive functions in e-sport that could be analyzed) to perform meta-analyse. 

  • Significant changes in our paper: due to the fact that we included more esports work, we had to broaden the search period. We also changed the title, we broadened the discussion and we clarified the purpose of the work.

Round 2

Reviewer 1 Report

Thank you for sending this back to me. This is a big improvement! I really appreciate the time and effort the authors have put in here to properly engage with esports literature. The new sections at the start of the paper are strong and comprehensive and very valuable, and the new sections at the end of the paper do a good job of combining the nutrition discussion with the specific demands and expectations of esports. This is a big improvement.

That said, I'd like to see two things before it's published. Firstly, I do think the central sections need to be integrated with the esports stuff a little more. I note the authors' comments that esports nutrition has not been studied and so they are hesitant to say "X might be good for Y"... but I do think the central section still reads too much like a literature review. I'd like to see a stronger integration of the two (like the authors have in the final discussion / conclusion parts!).

Secondly, the paper absolutely needs a close proofread. Although the quality of academic English is generally pretty high, lots of sentences have grammatical errors, extra words, omitted words, etc. This really needs a close proofread (this could also be from a professional proofreader) before it's ready to go out. Although the paper is generally understandable, it will make the paper a lot more readable if the language is polished before publication. As such, I highly recommend the authors do this.

But yes, this is a very strong improvement and I sincerely thank the authors for engaging so thoroughly with my feedback. I do still think the central section could be integrated into the esports stuff a bit more strongly - and the paper does need a good proofread - but this is a really strong improvement, and I'm very pleased to see it.

Author Response

Thank you very much for your feedback and further suggestions on our work in this article. At the outset, we would like to express our satisfaction with your opinion.

Please indicate in the text the fragments that require additional linguistic correction, because the previous version of the review was checked in this respect and highly rated in terms of language level. We would like to improve stylistic fragments of selected text in the editorial office.

The ingredient summaries were introduced in an extensive tutorial, so introducing the summarize to the center of the text would trigger repetition of text, which we would like to avoid. Therefore, at the stage of previous amendments, we described the chapters, respectively: eg. "The influence of dietary micronutrients on cognitive functions" (as a general overview) and later: "The potential role of minerals in esports".

Additionally, the following changes have been introduced:

- the word "also" has been introduced in the title to emphasize the doubts related to the limited amount of research available

- the word "potential" has been added in the discussion subchapters

- the words "esports" and "s-athletes" have been unified in the text

- we added the missing articles to the table - it escaped our attention at the previous stage

- we added additional information about the importance of micronutrients on cognitive functions.

Reviewer 3 Report

Again, there is a clear need to understand the relationship between nutrition and cognitive function for the athletes of e-sports, but there is not enough specific literature for a review.  While the edits are appreciated, they appear rushed.  The very first sentence has a syntax error and the new title is more misleading than before.  Asside from those points, the main issue stands (in which my rationale for rejection is based on).  I would recommend to back off of an e-sports title, introduction, and discussion until there are more specific studies on the exact topic.  Currently, the information summarized in your review is about specific nutrients and their impact on cognitive function and should be framed on that only, in my opinion.  A review on that topic would still be relevant to esports without making the review explicitly say it is about e-sports.  

Author Response

Thank you for your feedback and further suggestions. All the changes we have introduced so far were to highlight in the review what features are important in e-sport and what micronutrients and dietary supplements could potentially improve them. 

We 100% agree that there are currently no scientific reports on the influence of diet on cognitive functions in e-sport, which we emphasize many times in the review, but we indicate here the potential impact of micronutrients and dietary supplements on other functions based on scientific research conducted so far on other groups. So far, two papers have been prepared in which the influence of caffeine on the cognitive abilities of e-athletes has been studied. So we exclaim in the article that there is a need for further research. 

The change you suggested to remove "esports" from the content of the review would require us to expand the review with content related to aging, neurodegenerative diseases etc., which was not supposed to be the topic of our work.

In view of your job title change proposal, we decided to make a change and express some concerns about the nutrient enhancement effects in esports that should be investigated in the future. In addition, we decided that in the introduction and in the conclusions, we would mention the importance of cognitive functions in maintaining health at the beginning.

Language errors will be edited by the translator.

This manuscript is a resubmission of an earlier submission. The following is a list of the peer review reports and author responses from that submission.